# Brief communication: Sea-level projections, adaptation planning, and actionable science

William H. Lipscomb [1], David Behar [2], and Monica Ainhorn Morrison [1]

[1]Climate and Global Dynamics Laboratory, NSF National Center for Atmospheric Research, Boulder, CO, USA
[2]San Francisco Public Utilities Commission, San Francisco, CA, USA

**Correspondence:** William H. Lipscomb (lipscomb@ucar.edu)

**Abstract.**

As climate scientists seek to deliver actionable science for adaptation planning, there are risks in using novel results to inform decision-making. Premature acceptance may lead to maladaptation, practitioner confusion, and "whiplash". We propose that scientific claims should be considered actionable (i.e., sufficiently accepted to support near-term adaptation action) only after meeting a confidence threshold based on the strength of evidence as evaluated by a diverse group of scientific experts. We discuss an influential study that projected rapid sea-level rise from Antarctic ice-sheet retreat but in our view was not actionable. We recommend regular, transparent communications between scientists and practitioners to support the use of actionable science.

## 1 Introduction

Increasingly, climate research published in scientific journals is guiding adaptation planning and public discourse. Many communities are evaluating their vulnerability to climate change and identifying the actions needed to adapt. The stakes are high, since adaptation solutions can be expensive, politically difficult, socially inequitable, and ecologically disruptive. As sea levels rise, some communities may even be told to abandon their neighborhoods under "managed retreat". Meanwhile, scientific understanding of the impacts of climate change on humans and ecosystems is ever-changing, driven by advances in climate models, observations, and computing. Progress can be slow because of the complexity of interactions among the atmosphere, ocean, land, and cryosphere. Understanding comes in fits and starts, as replication of novel results leads to wider agreement. All predictions carry some uncertainty, which cannot always be quantified.

It is unsurprising, then, that interactions between scientists and practitioners[1] have been fraught with difficulties. For example, a recent global survey of practitioners (Hirschfeld et al., 2023) revealed widely varying approaches to identifying sea-level projections for planning, showing confusion in translating science into action. Practitioners are eager to use the best available

---

[1]We define practitioners as employees of governments and other entities that are responsible—legally or otherwise—for developing, implementing, and managing adaptation measures to protect people, infrastructure, assets, and communities. Practitioners include staff charged with evaluating and developing solutions to climate-related risks who work for local, state/provincial, and national governments, land managers, corporations, and other public or private sector entities. This term would also include policy advisors—those who analyze complex issues and develop options, given a defined policy. We distinguish practitioners from policymakers—the legislators and other government officials who create laws and regulations.

science, but not all published research is useful and relevant for adaptation planning and action. Acting hastily on novel claims can lead to costly maladaptation and policy chaos. Scientists can lose credibility by too frequently modifying the information sought by practitioners; this is known as the "whiplash effect" (Revkin, 2008).

Actionable science was first defined in the climate context in 2009 by members of the Water Utility Climate Alliance (WUCA) to explain to the scientific community the importance of distinguishing speculative claims in research literature from claims backed by robust evidence. They defined actionable science as "data, analysis, and forecasts that are sufficiently predictive, accepted, and understandable to support decision-making, including capital investment decision-making" (Behar, 2009). This term has been widely adopted in the science, government, and adaptation communities (e.g., U.S. Army Corps of Engineers, 2012; Executive Office of the President, 2013; WCRP Joint Scientific Committee (JSC), 2019). The WUCA definition hints at the kind of knowledge appropriate for driving adaptation action ("sufficiently accepted"), but does not guide practitioners in identifying this knowledge. Similarly, Cash et al. (2003) argued that scientific information can motivate action if it is seen as credible, salient, and legitimate, but they did not offer criteria for credibility.

Here, we will say that a claim is actionable when it is sufficiently accepted to justify adaptation action in the near term (assuming that other requirements for actionability, such as salience and legitimacy, have also been met). Near-term actions—for example, physical measures such as building seawalls and levees, as well as financial investments such as acquiring land—may be needed not only to address short-term vulnerability but also to prepare for long-term climate impacts. Thus, uncertainties about the rate of climate change in the next several decades (to 2100 and sometimes beyond) must be factored into near-term decisions. The longer the expected lifetime of an adaptation measure, the greater the uncertainty about how much the Earth system will change during its lifetime and whether expensive or disruptive measures are justified.

Researchers and practitioners have developed frameworks for decision making under deep uncertainty (DMDU) (Marchau et al., 2019a). Deep uncertainty arises when experts are unable to "specify the appropriate models to describe interactions among the system's variables, select the probability distributions to represent uncertainty about key parameters in the models, and/or value the desirability of alternative outcomes" (Lempert et al., 2003; Marchau et al., 2019b). DMDU frameworks support a shift from a "predict then act" paradigm to a "monitor and adapt" paradigm (Marchau et al., 2019b). In the new paradigm, the focus is on exploring a wide range of plausible futures and committing to short-term actions, while keeping open long-term options that might be triggered by new evidence. Ideally, costly actions are deferred until they are necessary. In practice, however, deeply uncertain claims can lead to inappropriate actions if not carefully communicated to practitioners.

Our goal is to guide scientists and practitioners in identifying scientific claims that are sufficiently accepted to be actionable. We hope to reduce the likelihood of financial, aesthetic, social, and environmental harm from the misinterpretation of novel scientific findings. We first discuss science and epistemology—the study of how we know what we know. Drawing on ideas from the philosophy of science, we describe the creation of knowledge as a community process of which peer review is only one element. We propose an epistemic criterion for actionable scientific claims. We then present a case study, describing sea-level projections that were communicated to practitioners in ways that led to their misuse. We conclude with recommendations for scientists and practitioners.

## 2 Science and epistemology

How do scientists create knowledge? Popular accounts sometimes focus on the contributions of lone geniuses. This is an inaccurate description of scientific practice, especially since the emergence of large-scale, publicly funded science in the second half of the 20[th] century. Furthermore, many philosophers of science—including Helen Longino, from whose work we draw here—have argued that the organization of scientists in communities leads to greater understanding and more reliable predictions. Longino (1990) emphasizes that scientific knowledge is social knowledge; it is created by many people working together through a "clashing and meshing of a variety of points of view" (p. 69). Publication in a peer-reviewed journal does not make an idea "a brick in the edifice of knowledge" (p. 69). The critical treatment of ideas after publication is equally important, because it allows other scientists to challenge background assumptions, assess how well the current evidence supports a given claim, and gather new evidence that can confirm or refute it. This critical treatment requires cognitive diversity—that is, a wide range of disciplinary backgrounds, research skills, and problem-solving strategies (Rolin et al., 2023). As different points of view are offered and heard, the community can sift out individual biases and reach a more objective consensus.

Longino asserts that this process is essential to building knowledge but is "de-emphasized in a context that rewards novelty and originality" (p. 80). In other words, the professional rewards for publishing bold claims often exceed the rewards for determining whether those claims are justified. She argues that "critical activities should receive equal or nearly equal weight to 'original research' in career advancement" (p. 76). Without critical review, novel claims may be used prematurely to support decisions.

Although space does not allow us to defend these arguments in detail, they are consistent with recent work in the philosophy of science (see, e.g., the overviews by Longino (2019) and Rolin et al. (2023)) and with our own experience of doing science, observing scientists, and using science in planning. We think Longino's analysis is especially relevant for climate science and adaptation planning.

The peer-reviewed journals that publish leading-edge climate research can be roughly divided into two groups. First are the disciplinary journals overseen by geoscience organizations such as the European Geosciences Union and the American Geophysical Union. Second are the "high-impact" multidisciplinary journals, including *Nature* and *Science*. Both sets of journals prioritize substantial, original research. There are greater professional rewards, however, for publishing in high-impact journals, which have wider audiences and favor work that is seen as novel and newsworthy. These journals are less likely to publish negative results—for example, the finding that a certain climate feedback is insignificant—even when the results are scientifically important and methodologically sound (Mehta, 2019).

When practitioners learn about climate research through media reports, they are likely to give disproportionate attention to a small number of studies in high-impact journals, focused on 21[st] century global-scale threats (Perga et al., 2023). Press releases from journals and universities often cast the work in a dramatic light, and media stories with attention-seeking headlines heighten the drama. This creates risks for practitioners. If they rely on media accounts to alert them to the "best available science", they may give undue weight to worst-case scenarios. And if they regard high-impact claims as immediately actionable, they short-circuit the critical process needed to transform novel claims into accepted knowledge.

For adaptation planning, the scientific assessments of the Intergovernmental Panel on Climate Change (IPCC) are more reliable than single studies. These assessments are directed mainly to policymakers but are read by practitioners with an eye toward what is actionable (Boyle et al., 2022). IPCC policies and procedures (IPCC, 2013) have evolved over several assessment cycles to describe the state of knowledge on climate science as accurately as possible. In particular, the IPCC assembles author teams with a range of scientific expertise, incorporating geographic diversity and gender balance. These teams carry out open, transparent reviews of all available literature. As a result, it is less likely that questionable assumptions will go unchallenged or pertinent evidence overlooked.

The IPCC has developed consistent ways to describe scientific uncertainty. The guidance note of Mastrandrea et al. (2010) sets forth two metrics for communicating uncertainty: (1) quantified measures, expressed probabilistically, and (2) "confidence in the validity of a finding", expressed using the qualitative descriptions "high", "medium", and "low". Confidence derives from the "type, amount, quality, and consistency of evidence (e.g., mechanistic understanding, theory, data, models, expert judgment) and the degree of agreement"[2]. Evidence is most robust, according to this guidance note, when there are "multiple, consistent independent lines of high-quality evidence"—e.g., evidence from a combination of global climate models, detailed process models, paleoclimate proxy data, and historical observations. To illustrate the importance of independent evidence, Winsberg (2018) gives the example of equilibrium climate sensitivity (ECS). Two climate models computing a similar ECS might not be independent; they might agree, for example, because they have similar but erroneous cloud feedbacks. In this case, paleoclimate data and instrumental records are valuable independent constraints, because their uncertainties are distinct from model uncertainties. If three independent methods agree that ECS is within a certain range, it is hard to explain why all three would have large errors in the same direction.

The IPCC guidance states that the presentation of low-confidence findings "should be reserved for areas of major concern, and the reasons for their presentation should be carefully explained". One reason to present a low-confidence claim might be that it has gained currency in the scientific community despite a lack of high-quality evidence. It is better to discuss such a claim, including the gaps in the evidence, than to disregard it. Also, low confidence is often a source of deep uncertainty, reflecting the inability of experts to agree on a modeling approach or assign meaningful probabilities to future events. Presenting low-confidence claims to practitioners can support stress testing and the design of adaptive pathways in DMDU frameworks, while signaling that acting immediately on such claims would be risky.

These philosophical accounts of knowledge creation, combined with IPCC practices, point the way toward an epistemic criterion for actionable science. We propose the following: *A scientific claim is sufficiently accepted to justify adaptation action (i.e., near-term physical measures and financial investments) when it is supported by multiple, consistent independent lines of high-quality evidence leading to high or medium confidence, as determined by a diverse group of experts in an open, transparent process.* This criterion reflects Longino's view that scientific knowledge is created by communities and that peer-

---

[2]Unlike Mastrandrea et al. (2010), we will refer to evidence but not agreement as a source of confidence. As Rehg and Staley (2017) have pointed out, "agreement" could refer either to a social consensus among scientists or the degree to which different research findings converge in supporting a scientific claim. These authors argued that social consensus is neither necessary nor sufficient for confidence and that in practice, the IPCC defines agreement in the second sense. For this reason, we have taken "consistency of evidence" to imply the kind of agreement leading to confidence, instead of treating agreement as separate from evidence.

reviewed claims must be scrutinized by a diverse group of scientists before they can be considered robust. It is based on existing practices and does not require scientists to learn new ways to assess the literature.

We urge that low-confidence claims be treated with caution. These claims can be used for planning and stress testing in DMDU frameworks (Marchau et al., 2019a), but should be treated differently from claims with wider support and should not determine near-term actions. Otherwise, adaptation practices will fluctuate as the science evolves, defeating the goal of having robust strategies over a range of possible futures (Lempert et al., 2003).

The boundary between planning and action can be fuzzy. For example, suppose that coastal practitioners are designing a levee with an expected lifetime of 75 years, based on a medium-confidence projection with a likely upper bound of 1.0 m of sea-level rise (SLR) by 2100, and a small but nonzero probability of 1.5 m. They might choose a design based on the 1.0 m projection, with an option for reinforcement if future evidence points toward a 1.5 m rise. If a new (low-confidence) paper claims that SLR could reach 3 m by 2100, and if preparing for 3 m has a similar cost to preparing for 1.5 m, then it may be sensible to revise the contingency plan to allow for further reinforcement. But if the new plan requires expanding the footprint for levee expansion, with a high near-term cost, then incorporating the low-confidence projection in the levee design probably would not be justified.

Next, we discuss an example of sea-level projections that were deemed actionable for practitioners in the absence of community confidence, resulting in confusion and whiplash.

## 3  Ice-sheet and sea-level projections

Global mean sea level (GMSL) is rising by about 3.7 mm yr$^{-1}$, mainly because of ocean thermal expansion and the loss of ice from the Greenland and Antarctic ice sheets (GrIS and AIS) and mountain glaciers (Fox-Kemper et al., 2021). Uncertainty in long-term sea-level projections is dominated by the AIS, which contains a large mass of ice that is grounded below sea level and is vulnerable to retreat under climate warming. If melted, this Antarctic ice could raise sea level by several meters.

The IPCC Fifth Assessment Report (AR5; Church et al., 2013) projected GMSL rise of 0.52–0.98 m by 2100 in a scenario with high greenhouse gas emissions (RCP8.5). The authors cautioned that the high end of this range was not an upper bound, because it excluded the possible collapse of marine-based sectors of the AIS. They cited the structured expert judgment (SEJ) study of Bamber and Aspinall (2013) (hereafter BA13), which found that "expert estimates of the contribution from this source have a wide spread, indicating a lack of consensus on the probability for such a collapse".

A few years later, DeConto and Pollard (2016) (hereafter DP16) published a study in *Nature* arguing that AIS mass loss alone could raise global sea level by more than 1 m before 2100 and more than 15 m by 2500. They were motivated by paleoclimate records showing that GMSL during the mid-Pliocene (3 million years ago, when temperatures were up to 3°C above present-day values) was 5–20 m higher than today (Fox-Kemper et al., 2021). They proposed mechanisms that could account for Pliocene SLR and possibly drive future SLR: atmospheric warming leading to hydrofracture of floating ice shelves, followed by the failure of marine-terminating ice cliffs (also known as marine ice cliff instability, or MICI). In simulations of

the past and future[3], they found large-scale, rapid ice loss in runs with MICI, but not without. In high-warming scenarios with MICI, the rate of GMSL rise due to West Antarctic Ice Sheet (WAIS) collapse reaches 20 mm yr$^{-1}$ by 2100 and more than 40 mm yr$^{-1}$ by 2150. In runs with ocean warming only—i.e., without strong atmospheric warming to drive MICI—the rate of WAIS retreat and GMSL rise is more than an order of magnitude lower.

The DP16 argument can be summarized as follows: Hydrofracture and MICI can explain large Pliocene SLR; therefore we can expect large, rapid SLR in a future climate that resembles the Pliocene. Several background assumptions are at work here: that other processes do not explain Pliocene SLR; that the DP16 ice sheet model accurately simulates hydrofracture and MICI; and that ice shelves could collapse before 2100 in a warming climate.

Each assumption has come under scrutiny. MICI has not been observed for present-day Antarctic ice shelves, and thus the collapse rate in the model is poorly constrained. Reese et al. (2018) argued that the model's treatment of the grounding line (the boundary between grounded and floating ice) is inaccurate for most ice shelves. Edwards et al. (2019) showed that for Pliocene SLR up to 15 m—within the uncertainty range—processes other than MICI could explain the paleo record. Bassis et al. (2021) developed a mechanistic model in which cliffs collapse catastrophically only over a restricted range of ice configurations. In a follow-up to DP16, DeConto et al. (2021) revised the atmospheric forcing, delaying hydrofracture and lowering the projected 21$^{st}$ century AIS sea-level contribution to 0.5 m or less, even if MICI is active. Recently, Morlighem et al. (2024) implemented a physically-based cliff parameterization in three different ice sheet models and found that ice acceleration and thinning tend to stabilize cliffs, even after several decades of forced grounding-line retreat. These results suggest that the WAIS is not vulnerable to MICI before 2100, although it is vulnerable in the longer term to marine ice sheet instability (MISI) (Seroussi et al., 2024).

In our view, the DP16 claims of rapid Antarctic ice loss and GMSL rise did not justify adaptation action, since they were not supported (either in real time or in retrospect) by multiple, independent lines of high-quality evidence. Nonetheless, DP16 has figured prominently in adaptation planning, as described in the next section.

## 4  Communicating sea-level projections to practitioners

DP16 has been highly influential. It received more news and social media attention in 2016 than any other climate paper published that year (McSweeney, 2017) and has been cited more than 1200 times in peer-reviewed journals. Initial media reports glossed over the scientific uncertainties, with dramatic headlines such as "Climate Catastrophe, Coming Even Sooner?" (Kolbert, 2016) and "Scientists nearly double sea level rise projections for 2100, because of Antarctica" (Dennis and Mooney, 2016). The latter headline is misleadingly ambiguous, since "Scientists" could refer either (correctly) to the authors of the paper or (incorrectly) to the broader scientific community. The *New York Times* compared the results to "the plot device of a Hollywood disaster movie" (Gillis, 2016). The general message was that a new peer-reviewed study had overturned the community consensus.

---

[3]In addition to the Pliocene, DP16 simulated the AIS during the Last Interglacial (LIG), when global mean air temperatures were similar to today and GMSL was about 6–9 m higher. Since the LIG atmosphere was too cool to trigger shelf collapse, these simulations required Southern Ocean warming of at least 3°C to drive WAIS retreat. DP16 suggested that ocean warming might have led to regional atmospheric warming that amplified the ice loss, but the atmosphere was not the primary driver.

Publications aimed at coastal adaptation planners highlighted the DP16 projections. Perhaps the most influential was Sweet et al. (2017) (hereafter S17), a multi-agency U.S. government report on future SLR. S17 sought to "support a wide range of assessment, planning, and decision-making processes", signaling the aim to influence practitioners. To be consistent with "recent updates to the peer-reviewed scientific literature", they issued an "Extreme" GMSL projection of 2.5 m by 2100 for RCP8.5, exceeding the previous upper bound of 2.0 m based on Pfeffer et al. (2008). To support this projection, S17 cited several studies, most prominently Kopp et al. (2014) (hereafter K14) and DP16. Based on process modeling and expert assessments and elicitation, K14 presented a very likely (90% probability) range of 0.5–1.2 m GMSL rise by 2100 under RCP8.5. By fitting a log-normal distribution to AR5 results and the BA13 Antarctic projections, they estimated a 0.1% probability of GMSL exceeding 2.45 m. This value was the source of the Extreme scenario. S17 stated further that the processes modeled by DP16 could "significantly increase the probability of the Intermediate-High, High, and Extreme scenarios"—i.e., that the likelihood of 2.5 m GMSL rise by 2100 might be much greater than 0.1%, because of MICI. By our criterion, the Extreme scenario was not actionable, since it was based on probabilities extrapolated from BA13 without reference to physical processes that were understood with at least medium confidence.

Projections based on DP16 and S17 were included in several reports commissioned by local and state jurisdictions for use by adaptation planners: "Climate Ready Boston" (Boston Research Advisory Group, 2016); "Rising Seas in California" (Griggs et al., 2017); and the "Unified Sea Level Rise Projection: Southeast Florida, 2019 Update" (Southeast Florida Regional Climate Change Compact Sea Level Rise Work Group (Compact), 2020). The Boston report relied on DP16—published just a few months earlier—for its "maximum possible" regional projection of 3.2 m SLR by 2100 under RCP 8.5. In the recent global survey of more than 250 jurisdictions (Hirschfeld et al., 2023), this was the highest figure recommended for planning. The California report recommended use of a 2100 high-end projection of 3.1 m for San Francisco, based on 2.5 m of GMSL rise (as in S17) plus regional factors driven by the DP16 Antarctic estimates. This projection was incorporated a year later in official guidance (California Ocean Protection Council, 2018); the high-end estimate was to be applied to any assets whose failure "would have considerable public health, public safety, or environmental impacts". The Southeast Florida report did not cite DP16 but relied on S17 to develop its high-end, 2100 SLR projection of 2.61 m, calling S17 "a reliable source of data from the national effort on sea level rise projections".

Recent community assessments have given less prominence to DP16 in light of subsequent research. The IPCC Sixth Assessment Report (AR6; Fox-Kemper et al., 2021) presented one set of high-end sea-level projections based on models with physical processes that are understood with at least medium confidence (e.g., Levermann et al., 2020; Edwards et al., 2021), and another set including low-confidence processes such as MICI (Bamber et al., 2019; DeConto et al., 2021). By offering two sets of projections, the IPCC aimed to alert readers to uncertainty and ambiguity without overshadowing projections for which there is more confidence (Kopp et al., 2023). AR6 classified MICI as a deeply uncertain, low-confidence process because there is no accepted theory of its exact mechanism and limited evidence that MICI has taken place in the past or present. Similarly, AR6 ascribed low confidence to the SEJ study of Bamber et al. (2019) because it was unknown how many experts included low-confidence processes in their estimates. The low- and medium-confidence projections differ considerably. Under a high-emissions scenario (SSP5-8.5), the 95[th] percentile upper bounds of GMSL by 2100 are 1.6 m and 2.4 m for medium-

confidence and low-confidence processes, respectively. The gap between medium and low confidence widens by 2150, with respective 83$^{rd}$ percentile upper bounds of 1.9 m and 4.8 m.

Sweet et al. (2022) (hereafter S22), the successor report to S17, removed the Extreme projection of 2.5 m because it is now "less plausible". This prompted backtracking in the revised California guidance (California Sea Level Rise Guidance, 2024). S22 retained a High projection of 2.0 m, which includes a large Antarctic contribution based mainly on Bamber et al. (2019) and DeConto et al. (2021). In our view, the High projection is not actionable because it relies heavily on low-confidence science. S22 also kept an Intermediate-High scenario of 1.5 m, in which "deeply uncertain ice sheet processes play important roles" late in the century. Since S22 did not quantify the role of low-confidence processes in a transparent way, it would be difficult to say whether the Intermediate–High scenario is actionable, without appealing to medium-confidence projections of similar magnitude (e.g., in AR6). A clear separation of medium- and low-confidence science, as in AR6, would have been more useful for practitioners.

AR6 and S22 did not assign likelihoods to deeply uncertain mechanisms such as MICI, implying that the likelihood attached to the low-confidence projections is unknown. The study of van de Wal et al. (2022) (hereafter V22) made a stronger claim: that under 5°C of warming, as might occur under SSP5-8.5, there are no physically plausible processes, including MICI, that would raise sea level by more than 1.27–1.55 m before 2100. The 1.55 m value is the sum of contributions from thermal expansion (0.36 m), glaciers (0.27 m), the GrIS (0.29 m), the AIS (0.59 m), and land water storage (0.04 m), assuming perfect correlation between all contributions, while the lower value assumes independent contributions. These estimates were based on a variety of physical arguments, including a judgment that large ice shelves are unlikely to collapse during this century, and therefore widespread MICI before 2100 is implausible. The authors drew on multiple lines of evidence suggesting medium confidence.

The V22 projections, combined with recent model simulations showing at most a minor role for MICI through 2100 (Morlighem et al., 2024), suggests an emerging scientific consensus that GMSL rise of more than 1.5 m by 2100 can be ruled out, at least with medium confidence. Thus, the highest estimates (∼3 m) of a few years ago were twice as high as current knowledge would justify. Lempert et al. (2024) noted that AR5 and AR6 "opened the aperture" to provide a wider range of possible futures by discussing low-confidence science in detail. Consistent with V22, we argue that practitioners should embrace widening uncertainty in designing adaptation action only when justified by scientific confidence.

Figure 1 shows the recent history of high-end sea-level projections from scientific community assessments and practitioner-oriented reports. "High-end" does not have the same meaning in all cases, but typically refers to scenarios judged to be physically plausible (though not likely) and/or with an estimated probability of ∼1–5%, assuming high emissions and warming (e.g., RCP8.5 or SSP5-8.5). The projections are shown in three groups: (1) assessments and reports published during 2012–2016, (2) S17 and other reports published in 2016–2020, soon after DP16, and (3) reports and assessments since 2020. Group 2 stands out with projections of well over 2 m, which were excursions from a relatively stable base of projections of 2 m or less. For example, the two broad surveys by Horton et al. (2014, 2020) gave similar high-end projections of 1.5 m and 1.65 m. The lower projections in Group 3 reflect the removal of DP16-based MICI contributions after community review, along with a desire (in V22) to reduce chaos for practitioners by basing high-end estimates on physical plausibility supported by multiple lines of evidence. Within Group 3, the highest values (2.0 m) are from S22 and California Sea Level Rise Guidance (2024),

which include a reduced but still sizable MICI contribution based on DeConto et al. (2021). The three lower values, clustered around 1.6 m, exclude low-confidence processes. By our criterion, only the lower projections in Group 3 are actionable.

We suggest that California coastal planners have been ill-served by shifting targets in state guidance: from 1.4 m in 2013 to 3.1 m in 2018 and back to 2.0 m in 2024 (Fig. 1). The main reason for this case of whiplash was the decision in 2018 to adopt low-confidence high-end projections. Consistent application of actionable science criteria would have migrated high-end projections from 1.4 m to about 1.5 m over those 12 years, reflecting a more constrained reading of the scientific landscape based on practitioner needs.

Finally, we would like to give two examples of practitioner guidance that avoided the pitfalls described above. First, the most recent sea-level guidance from the Met Office Hadley Centre (Palmer et al., 2018) retained high-end projections adopted in 2009. Citing DP16, the authors noted that "marine ice cliff instability has been proposed as an important potential feedback" but added that "further research is required to strengthen the observational evidence for, and prevalence of, this mechanism". Second, the 2017 New Zealand coastal adaptation guidance (Ministry for the Environment, 2017) proposed a high-end ("H+") scenario of 1.05 m GMSL by 2100, based on AR5, as part of a dynamic adaptive pathways planning (DAPP) strategy. The updated guidance (Ministry for the Environment, 2024) retained the DAPP approach and used the AR6 medium-confidence projections for SSP5-8.5 to design an H+ scenario with 1.1 m GMSL by 2100. The new projection was described as a "plausible upper range" for SLR and was recommended for "high-end stress testing of adaptation options and pathways". The AR6 low-confidence projections were assigned a limited role for "further stress testing" related to long-lived coastal development and managed-retreat options. Since the core recommendations in the New Zealand guidance have not relied on low-confidence science, there has been no whiplash.

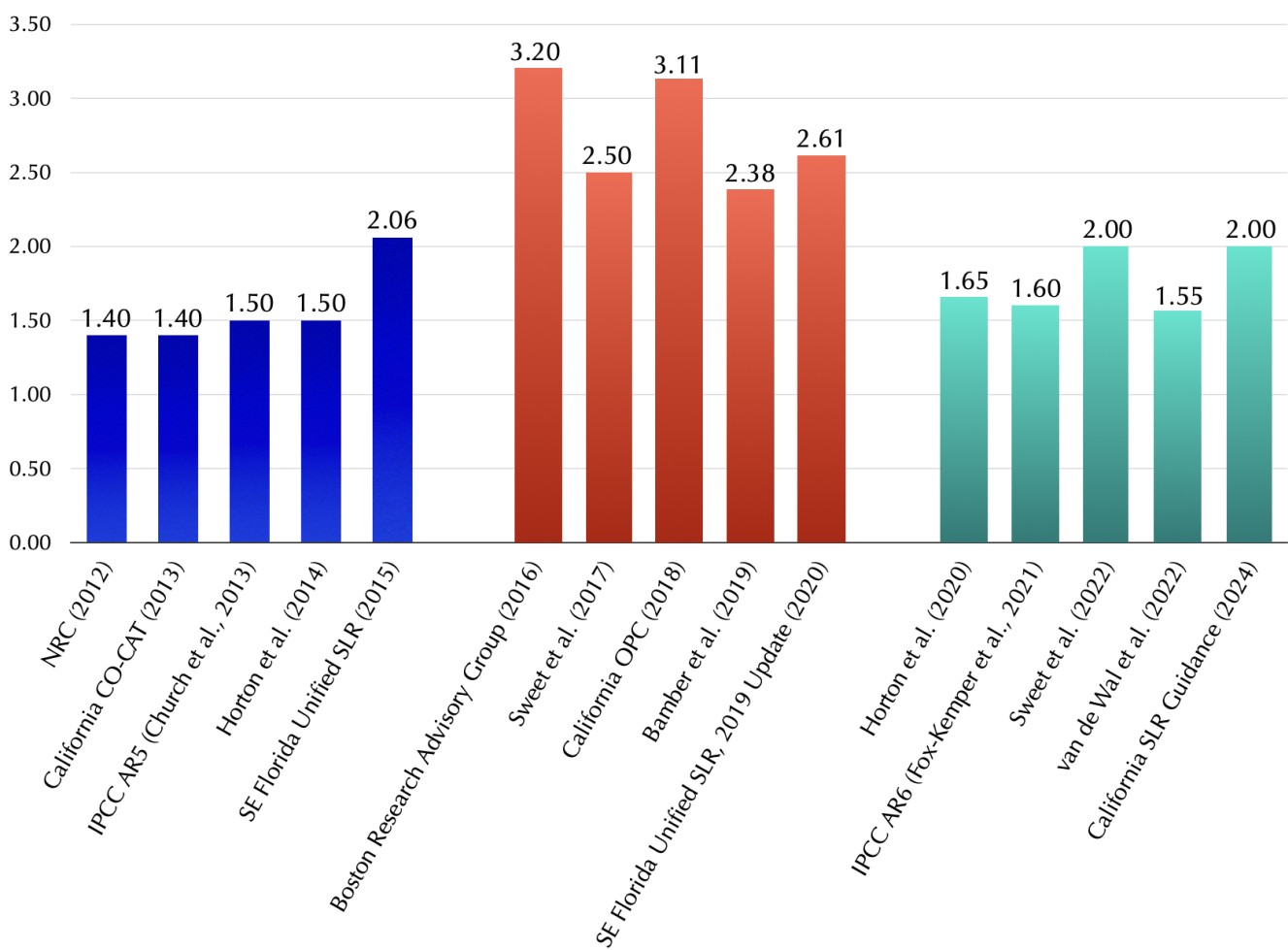

**Figure 1.** High-end projections for sea-level rise (m) by 2100, arranged chronologically: (1) community assessments and practitioner-oriented reports published before 2016 (blue), (2) reports and assessments published in 2016–2020 (red), and (3) recent reports and assessments (green). The recent reports gave less weight to DP16 and MICI than the 2016–2020 reports; Sweet et al. (2022) and California Sea Level Rise Guidance (2024) included MICI but at a reduced rate, while the other three recent assessments excluded MICI. Most projections are for global sea-level rise, although some jurisdictional reports (e.g., the 2016 Boston and 2018 California reports) include regional effects. The California OPC (2018) value is for the San Francisco tide gauge. The AR6 value is the 95th percentile upper bound based on medium-confidence processes. The van de Wal et al. (2022) value is the upper value in their high-end range, assuming perfect correlation between components. Baseline dates vary but typically are around the year 2000. All reports are cited in the text except for National Research Council (2012), Coastal and Ocean Working Group of the California Climate Action Team (CO-CAT) (2013), and Southeast Florida Regional Climate Change Compact Sea Level Rise Work Group (Compact) (2015).

## 5    Recommendations

We have proposed an epistemic criterion for actionable science: *A scientific claim is sufficiently accepted to justify adaptation action (i.e., near-term physical measures and financial investments) when it is supported by multiple, consistent independent lines of high-quality evidence leading to high or medium confidence, as determined by a diverse group of experts in an open, transparent process.* This criterion is informed by IPCC practices, by philosophical arguments that scientific knowledge is social knowledge, and by the DP16 case study.

We recommend that practitioners view novel peer-reviewed claims with caution, especially those that challenge the scientific consensus. They should be aware of these claims but not treat them as actionable before they are evaluated by the scientific community. Practitioners can reduce the risk of maladaptation and whiplash through careful reading of IPCC reports and other community assessments, focusing on findings that are affirmed with confidence after critical review. If incorporating low-confidence claims in planning, they should use decision-making frameworks designed to deal with deep uncertainty, so as not to act prematurely based on speculative science that is likely to evolve. Since IPCC reports are infrequent, we recommend new structures that regularly bring together scientific experts to assess ongoing research on sea-level rise and other fast-evolving topics.

We think that scientists have a professional duty, when presenting new results, to put them in the context of well-established science and to acknowledge uncertainties. When reporting on new results, journalists should seek a range of opinions to identify what remains unsettled. Journalists should also be mindful that not all breakthroughs are communicated through single studies in high-impact journals with bold press releases. Some breakthroughs emerge from multiple independent studies that support one another. Others are not evident at the time of publication, but only in hindsight with community corroboration of novel results.

We seek to make the sea-level assessment process as objective as possible, insulated from social or political pressure to make projections that are higher or lower than justified by scientific knowledge. There may be pressures to reduce sea-level projections to minimize adaptation costs and political difficulties. We have also observed pressures to adopt extremely high projections, perhaps to motivate mitigation action or get practitioner attention. Divorcing the development of actionable science from both of these dynamics can prevent multiple risks, including maladaptation, loss of scientific credibility, and undermining of the adaptation enterprise.

We recommend that scientists and practitioners work together to better manage the boundary between research and decision-making. Regular and intentional communication between these groups can reduce confusion and minimize the risk of maladaptation. Organizations such as the Practitioner Exchange for Effective Response to Sea Level Rise (PEERS), formed in part to create practice-centered collaboration with a diverse group of scientific experts, can promote understanding of actionable science where that understanding is most needed. There is a careful dance between research and practice that can succeed with clear ground rules and open communication.

*Author contributions.* All authors contributed to discussing the ideas and writing the manuscript.

*Competing interests.* The authors declare no competing interests.

*Acknowledgements.* WHL and MAM were supported by the NSF National Center for Atmospheric Research, which is a major facility sponsored by the National Science Foundation under Cooperative Agreement No. 1852977. We thank Michael Mastrandrea for helpful discussions during the early writing phase. We are grateful to Florence Colleoni, Rebecca Priestley, Chris Weaver, and one anonymous reviewer for their constructive comments. We also thank Jeremy Bassis, Rajashree Datta, Marjolijn Haasnoot, Robert Kopp, Robert Lampert, Judy Lawrence, and other colleagues in the climate modeling and practitioner communities for their reflections and analysis.

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
