# Peer review of "Brief communication: Sea-level projections, adaptation planning, and actionable science"

_EGUsphere, 2024_

## Author Comment (AC1)

**Reply to Christopher Weaver**

We thank the reviewer, Christopher Weaver, for his thoughtful comments on our manuscript. Below, his comments are in black font and our replies are in blue.
* * *
I appreciate the opportunity to comment, since I would like to point out, and help correct, an error in the paper.

Specifically, on page 6, the authors have made the following statement about the influence of DeConto and Pollard (2016) on the U.S. interagency sea level rise scenarios report of Sweet et al. (2017) (my emphasis):

"To be consistent with "recent updates to the peer-reviewed scientific literature", they issued an "Extreme" global mean sea-level projection of 2.5 m by 2100 for RCP8.5, exceeding the previous upper bound of 2.0 m based on Pfeffer et al. (2008). Their Extreme projection relied on a large AIS contribution, **based primarily on DP16**. [Footnote 4]

[Footnote 4] The 2.0 m upper bound of Pfeffer et al. (2008) assumed large contributions from both ice sheets: 0.54 m from the GrIS and 0.62 m from the AIS. In AR5, Church et al. (2013) estimated a likely upper bound of just 0.21 m for the GrIS, since process models do not support "the order of magnitude increase in flow" in Pfeffer et al. (2008). To reach an upper bound of 2.0 m or more, S17 therefore needed an increased AIS contribution of ~1.0 m or more. To support this increase, they cited DP16 along with the expert-judgment assessment of Bamber and Aspinall (2013). However, the latter study gave a high-end (95th percentile) estimate of 0.84 m SLR from the two ice sheets, less than Pfeffer et al. (2008). Other studies cited by S17 did not give independent evidence of a large AIS contribution. Thus, both the "High" projection of 2.0 m and the "Extreme" projection of 2.5 m in S17 relied on DP16's claim that the AIS could contribute at least a meter of SLR by 2100."

The highlighted statement is a misstatement of fact. The accompanying footnote is also erroneous, as well as the part of the statement, on page 7, that references DeConto and Pollard (2016) in the context of Sweet et al. (2017), i.e., "(2) S17 and other reports that were published in 2016–2020 **and relied on DP16 for the AIS contribution**."

Here is the relevant paragraph from page 14 in Sweet et al. (2017):

"The growing evidence of accelerated ice loss from Antarctica and Greenland only strengthens an argument for considering worst-case scenarios in coastal risk management. Miller et al. (2013) and Kopp et al. (2014) discuss several lines of arguments that support a plausible worst-case GMSL rise scenario in the range of 2.0 m to 2.7 m by 2100: (1) The Pfeffer et al.

(2008) worst-case scenario assumes a 30-cm GMSL contribution from thermal expansion. However, Sriver et al. (2012) find a physically plausible upper bound from thermal expansion exceeding 50 cm (an additional ~20-cm increase). (2) The ~60 cm maximum contribution by 2100 from Antarctica in Pfeffer et al. (2008) could be exceeded by ~30 cm, assuming the 95th percentile for Antarctic melt rate (~22 mm/year) of the Bamber and Aspinall (2013) expert elicitation study is achieved by 2100 through a linear growth in melt rate. (3) The Pfeffer et al. (2008) study did not include the possibility of a net decrease in land-water storage due to groundwater withdrawal; Church et al. (2013) find a likely land-water storage contribution to 21st century GMSL rise of -1cm to +11 cm. Thus, to ensure consistency with the growing number of studies supporting upper GMSL bounds exceeding Pfeffer et al. (2008)'s estimate of 2.0 m by 2100 (Sriver et al., 2012; Bamber and Aspinall, 2013; Miller et al., 2013; Rohling et al., 2013; Jevrejeva et al., 2014; Grinsted et al., 2015; Jackson and Jevrejeva, 2016; Kopp et al., 2014) and the potential for continued acceleration of mass loss and associated additional rise contributions now being modeled for Antarctica (e.g., DeConto and Pollard, 2016), this report recommends a revised worst-case (Extreme) GMSL rise scenario of 2.5 m by 2100."

In developing the report, we wished to provide a number of scenarios that could be used to fully bracket the evidence base for the physically possible 21st century sea level rise, as well as providing expert judgment about the central tendency/best guess trajectory. These ranged from 0.3 m at the lowest of the lower bounds to 2.5 m at the uppermost of the upper bounds. Briefly, the motivation was to support as wide a possible range of decision contexts as existed at the time in coastal risk planning and management (e.g., see Hinkel et al., 2015, Nature Climate Change, and many others), including long-term adaptation pathways approaches and "stress test" type applications, both of which often use a "not-to-be-exceeded," upper bound metric of performance.

As described in the quoted paragraph from Sweet et al. (2017), above, we arrived at the 2.5 m upper bound by synthesizing a number of lines of evidence from numerous studies, as well as the IPCC AR5, to individually interrogate the physically possible ranges of the contributing components to global-mean sea level rise. This was new evidence, and/or new synthesis of that evidence, since Pfeffer et al. (2008), the study that helped define the physically possible upper bound for a preceding U.S. interagency sea level rise scenarios report (Parris et al., 2012).

All of these studies predated the publication of DeConto and Pollard (2016); we had already decided on the 2.5 m upper bound, and completed most of the work of developing the global and regional scenarios, before that paper was published. As just one example, Kopp et al. (2014) estimated 2.45 m as the 99.9 percentile outcome for global-mean sea level rise in 2100 under RCP8.5. Once DeConto and Pollard (2016) was published, we added it to our citation list as another piece of evidence, but the conclusions of that paper had no influence on our choice of 2.5 m as the upper bounding scenario. The successor report to Sweet et al. (2017), i.e., Sweet et al. (2022), stated this clearly, as well (e.g., see page 11): "In Sweet et al. (2017), these scenarios

were developed to span a range of 21st-century GMSL rise from 0.3 m to 2.5 m. Sweet et al. (2017) built these scenarios upon the probabilistic emissions scenario–driven projections of Kopp et al. (2014)."

The bottom line is that, if DeConto and Pollard (2016) had never been published, we would have written exactly the same report at the time that we wrote it.

We thank the reviewer (CW) for clarifying the reasoning that led to the 2.5 m "Extreme" projection in Sweet et al. (2017; hereafter S17). As he requests, we will remove language "stating or implying a reliance of Sweet et al. (2017) on DeConto and Pollard (2016)".

We would also like to respond to the statement that "if DeConto and Pollard (2016) had never been published, we would have written exactly the same report at the time that we wrote it". We think that DP16 plays an important role in the main arguments of S17, even if it was not the source of the 2.5 m projection. S17 cited DP16 nine times by our count, often in support of important claims. For example:

- S17 cited DP16 (p. 3) to support the following statement: "Sea level science has advanced significantly over the last few years, especially improving understanding of the complex behaviors of the large, land-based ice sheets in Greenland and Antarctica under global warming, and the correspondingly larger range of possible 21$^{st}$ century rise in GMSL than previously thought." The Extreme projection relies on the long tail in Kopp et al. (2014; hereafter K14), which was based on the expert elicitation study of Bamber and Aspinall (2013; hereafter BA13) but did not specify mechanisms that could make the long tail physically possible. By exploring the mechanisms of hydrofracture and MICI, DP16 aimed to provide a physical foundation for a large Antarctic sea-level contribution. To the extent that the mechanisms described in DP16 were thought to be plausible, the Extreme projection was more credible to practitioners than would have been the case based on BA13 and K14 alone.
- S17 stated (p. 13) that "additional GMSL rise upwards of 0.6–1.1 m to median estimates under RCP8.5 are possible by 2100 (DeConto and Pollard, 2016), potentially raising median GMSL projections for RCP8.5 of Kopp et al. (2014) as high as 1.9 m by 2100." Without DP16, a practitioner reading S17 might have minimized the relevance of the 2.5 m projection for decision-making, given the low 0.1% probability. However, the addition of ~1 m to median estimates would suggest that under RCP8.5, the Extreme scenario is much more likely than 0.1% and therefore should be taken seriously in planning.
- The last sentence in the paragraph quoted by CW (p. 14) cites DP16—with its "potential for continued acceleration of mass loss and associated additional rise contributions now being modeled for Antarctica" —in support of the Extreme scenario.
- S17 stated (p. 21) that "as discussed in Section 3, new evidence regarding the Antarctic ice sheet, if sustained, may significantly increase the probability of the

Intermediate-High, High, and Extreme scenarios, particularly for RCP8.5 projections based upon Kopp et al. (2014)." The context indicates that "new evidence" refers to DP16.

- Figure 8 (p. 22) shows the study's six representative SLR scenarios in relation to historical GMSL reconstructions. The Extreme scenario is shown with a red curve that reaches 2.5 m in 2100. To the right side of the graph, three boxes illustrate the 5th–95th percentile ranges of RCP-based GMSL projections from recent studies. The range is 0.5–1.3 m for the RCP8.5 scenario, shown in red. Appended to these boxes are dashed lines described as "the difference between the median Antarctic contribution of the Kopp et al. (2014) probabilistic GMSL/RSL study and the median Antarctic projections of DeConto and Pollard (2016)". For RCP8.5, the red dashed line representing the DP16 contribution extends from 1.3 m to about 2.4 m, i.e. nearly to the top of the Extreme curve. Thus, the figure suggests that the DP16 Antarctic projections for RCP8.5 are able to bridge the gap between the 95th percentile RCP8.5 projection and the Extreme scenario, lifting the probability of the Extreme scenario from 0.1% (the value from K14) to a value many times greater, perhaps ~5%.

Thus, S17 without DP16 would not have been "exactly the same report" and likely would not have been as influential for adaptation planners. As our manuscript states, the reports developed by practitioners and practitioner advisors in the science community (e.g., Boston Research Advisory Group, 2016; California OPC, 2018; and the Griggs et al. "Rising Seas in California" 2017 report underpinning the OPC report) refer prominently to DP16 as a key driver (perhaps *the* key driver) of high-end projections recommended for planning. We would submit that the signal sent by the multiple citations of DP16 in S17 was highly influential.

For example, Griggs et al. (2017) highlighted DP16 as the most important driver of the 3.1 m high-end estimate later adopted in California OPC (2018), while also pointing out the many unanswered questions associated with DP16 in an appendix substantially devoted to these uncertainties. This focus on DP16 became concrete when the OPC (2018) report instructed practitioners to use the 3.1 m "H++" projection in planning for any project that "would have considerable public health, public safety, or environmental impacts should this level of sea-level rise occur." A core goal of our Commentary is to advocate for a stronger underpinning for actionable science, as regulators and practitioners struggle to adjust to the realization that the high-end estimates in OPC (2018) were based on what is now understood to be low-confidence science.

In closing, I wanted to note that, on the initiative of one of the authors of this brief communication (DB), he and a number of others of us (including myself and Kopp, as well as DeConto) spent substantial time in productive discussions of the very points I have just summarized, and related topics, in the broader context of the nuances of using cutting-edge sea level rise science to support decision-making. These extensive discussions following the

publication of Sweet et al. (2017) resulted in an AGU presentation in December 2017 by DB (see https://par.nsf.gov/servlets/purl/10066643), and a written summary of our engagement (see https://acwi.gov/climate_wkg/minutes/nal_agu_consensus_statement_probabilisitic_projections_dec_2017.pdf), both of which reflected a useful integration of our diversity of perspectives as scientists and practitioners.

DB confirms that he led a group process including himself, CW, Robert Kopp, Rob DeConto, representatives from the US Army Corps, and others. We suggest that interested readers access this document at https://www.wucaonline.org/assets/pdf/pubs-sfpuc-agu-consensus-statement.pdf. (Neither location provided by CW appears to link to this document.)

The outcome of this process, "Consensus Statements: Planning for Sea Level Rise: An AGU Talk in the Form of a Co-Production Experiment Exploring Recent Science" (Behar et al., 2017; hereafter Consensus Statements) reports two goals for the process. The first was to address the increased appearance of Bayesian probabilistic projections intended for practitioner use, particularly K14, in documents intended for use by practitioners developing plans to address rising seas. Research led by DB indicated confusion among practitioners about the nature and meaning of Bayseian probabilistic projections. The authors of the Consensus Statements observed that these estimates "in many instances. . . are arriving on the desks of planners, engineers, and decision makers who have little background in the methodologies used…and do not provide sufficient guidance on how to use them in planning, decision making, or adaptation design context." While the Consensus Statements list a number of opportunities and limitations associated with Bayesian probabilities, it is worth repeating one that relates to the conversation here (emphasis added): "*There is no consensus on how to meaningfully assign quantitative probabilities for the upper extreme range of potential future global SLR*; therefore, a given set of Bayesian probabilistic projects may underestimate or overestimate the SLR contributions due to rapid ice sheet loss after 2050." The Consensus Statements go on to recommend that, to properly represent uncertainty, multiple analyses and PDFs, rather than a single Bayesian PDF, should underpin adaptation planning.

We think it was unfortunate that the California OPC (2018) guidance included the following statement: "Probabilistic projections represent consensus on the best available science for sea-level rise projections through 2150."  This statement is neither true nor consistent with the Consensus Statements drafted by DB, CW, and Dr. Kopp, among others.

The second goal of this group was to address DP16 which, according to the Consensus Statements, "suggested the potential for significantly higher upper end projections for Antarctic ice sheet melt, which increase both global and regional SLR above most previously assumed upper limits." However, the group did not achieve this goal. "The group did not completely fulfill one of its two objectives," the Consensus Statements said, "the consideration of how

DeConto and Pollard (2016), as a defining example of cutting-edge science leading to new upper end SLR estimates, can or should be incorporated into planning." Considering the chaos and confusion prevalent in the uptake of high-end projections into planning, as reported in our submitted manuscript and other sources (e.g., Stammer et al., 2019; Boyle et al 2022; van de Wal et al., 2022; Hirschfeld et al., 2023; Hirschfeld et al., 2024), DB wishes to express his regret that he failed to lead the authors of the Consensus Statements into this next round of conversations in 2018. Perhaps the authors, working together, could have helped mitigate the confusion that persists today about high-end projections, including how to treat modeling studies that project catastrophic Antarctic ice melt on an adaptation time scale but are not widely accepted in the science community (including DP16 and DeConto et al., 2021).

The authors should remove language stating or implying a reliance of Sweet et al. (2017) on DeConto and Pollard (2016). That would be a good first step in helping the paper be considered for publication.

We will remove this language.

Note that I do not, in any way, have any objection to the authors disagreeing with the decision in Sweet et al. (2017) to use 2.5 m globally by 2100 as the top-end, bounding scenario on other grounds. Such a disagreement would simply have to be justified in terms of the totality of references and lines of evidence summarized above, absent any reliance on DeConto and Pollard, as well as the stated purpose of the use of a limiting upper-bound scenario in that report - in other words, the choice to include 2.5 m not because it is at all likely, but precisely because it is very, very unlikely.

We do, in fact, disagree with the decision in S17 to use 2.5 m as the top-end global scenario. To support our disagreement, we will apply our actionable science criterion to the other studies cited in S17, not including DP16. S17 cited eight papers as among the "growing number of studies supporting upper GMSL bounds exceeding Pfeffer et al. (2008)'s estimate of 2.0 m by 2100": BA13, K14, Sriver et al. (2012), Rohling et al. (2013), Jevrejeva et al. (2014), Grinsted et al. (2015), Jackson and Jevrejeva (2016), and Miller et al. (2013). We will comment on each study, starting with BA13 and K14.

Table A1.1 in Sweet et al. (2022; hereafter S22) states that the Antarctic projections in S17 are based on the "*likely* range from IPCC AR5", with the "shape of tails" for high-end projections based on the structured expert judgment (SEJ) study of BA13, as interpreted by K14. BA13 gave a 95th percentile estimate of 0.84 m for the Greenland and Antarctic ice sheets together. K14 combined BA13 with independent estimates for glaciers and ice caps, thermal expansion, and land water storage to obtain a 95% upper bound of 1.21 m. This upper bound was based mainly on processes that were understood at the time with at least medium confidence. To derive their 99.5% and 99.9% upper bounds (1.76 m and 2.45 m, respectively), K14 had to assume a mathematical form for the tail probabilities. The Supporting Information in K14 states: "To

reconcile the AR5 and BA13 projections of ice sheet mass loss, we first fit log-normal distributions to the rates of ice mass change in 2100 for AR5 and BA13." They created hybrid curves (see their Fig. S1) which were scaled to match the median and likely ranges of AR5, with tails based on a log-normal fit to BA13. They did not try to justify the tails in terms of physical processes.

We do not think the statistical analysis in K14 was robust enough to underpin decision-making for adaptation planners. We refer to the quotations above from the Consensus Statements, in particular the statement that "there is no consensus on how to meaningfully assign quantitative probabilities for the upper extreme range of potential future global SLR". We reiterate that more robust efforts to display the significance and sources underpinning deep tails in Bayesian probabilistic projections would improve clarity for practitioners who are considering these outputs for adaptation planning.

Next, we will comment briefly on the other six studies.

Sriver et al. (2012) proposed an upper bound of 0.55 m for the thermal expansion (TE) contribution to GMSL. This estimate was based on a perturbed physics ensemble applied to an Earth system model of intermediate complexity (the UVic model) with a coarse-resolution ocean component. AR5 cited this paper but gave a likely range of 0.21–0.33 m for TE under RCP8.5, adding that "we have *high confidence* in the projections of thermal expansion using AOGCMs" (p. 1151; emphasis in original). Similarly, AR6 gave an upper bound of 0.36 m for TE. We conclude that the value of 0.55 m from Sriver et al. (2012) was an outlier based on a single model. This high projection was discounted by the AR5 authors prior to S17 and thus was not appropriate for use in adaptation planning.

Rohling et al. (2013) used the geologic record to inform projections of future SLR. They concluded that the geologic context supports SLR of up to 1.8 m by 2100 at 95% confidence. This estimate was based on Monte Carlo–style sampling of the distributions of three parameters in a logistic equation (their Eq. 1). They cautioned that their high-end estimate requires SLR rates approaching 4 m/century, similar to those associated with Meltwater Pulse 1a during the collapse of large Northern Hemisphere ice sheets about 14,000 years ago. This collapse might not be an appropriate analog for the future (since these ice sheets no longer exist) and in any case does not yield a projection as large as 2.5 m.

Jevrejeva et al. (2014), like K14, took the AR5 likely range as a starting point and used BA13 to estimate the additional ice-sheet contribution. They obtained a 95% upper bound of 1.8 m GMSL rise by 2100—well below 2.5 m. They noted that "large uncertainties remain due to the lack of scenario-dependent projections from ice sheet dynamical models, particularly for mass loss from marine-based fast flowing outlet glaciers in Antarctica. This leads to an intrinsically hard to quantify fat tail in the probability distribution for global mean sea level rise." The studies of Grinsted et al. (2015) and Jackson and Jevrejeva (2016) used similar methods and were broadly

consistent with Jevrejeva et al. (2014). None of these studies supports a 2.5 m projection. Moreover, these studies do not provide evidence independent of K14. Like K14, they rely on BA13, but they make different statistical inferences about the high end.

The only one of these six studies with a high-end GMSL projection exceeding 2.0 m is Miller et al. (2013). Starting from a projection of 2.0 m based on Pfeffer et al. (2008), these authors argued for additions of 0.1 m for land water storage (which Pfeffer et al. (2008) neglected), 0.25 m for TE based on Sriver et al. (2012), and 0.3 m for the Antarctic ice sheet based on BA13. We would challenge this projection on the following grounds:

- Pfeffer et al. (2008), which was taken as a starting point, assumed unrealistically high GrIS discharge. AR5 (which appeared after Miller et al. (2013) was submitted for publication) is a better starting point since it includes the land water storage term and has a much lower GrIS contribution.
- Sriver et al. (2012), as discussed above, is an outlier. It does not provide robust evidence for increasing the thermal expansion estimate.
- Miller et al. (2013) assumed an Antarctic contribution of 22 mm/yr by 2100 based on BA13. This is larger than the 95[th] percentile upper bound of 17.6 mm/yr in BA13 for the GrIS and AIS *combined*. To obtain a much higher Antarctic value than BA13, Miller et al. (2013) assumed perfect correlation of the estimated 95[th] percentile contributions from East and West Antarctica, without explaining why this assumption was justified. In our view, BA13 does not support a 30-cm increase (from 0.62 m to 0.94 m) for the AIS relative to Pfeffer et al. (2008).

We have proposed that actionable science should rest on multiple, consistent lines of high-quality evidence, resulting in medium or high confidence as evaluated by a group of experts in a transparent process. The eight studies above do not meet this criterion. Several of them (Miller et al., 2013; K14, Jevrejeva et al., 2014; Grinsted et al., 2015; Jackson and Jevrejeva, 2016) depend on BA13. In agreement with AR6, we would argue that SEJ studies like BA13 and Bamber et al. (2019) do not meet a medium-confidence threshold, since the expert surveys can incorporate low-confidence processes in a non-transparent way. Sriver et al. (2012) is an independent line of evidence, but as early as 2013, this evidence was assessed as not being of high quality. Rohling et al. (2013) added evidence from the geologic record but did not support a projection above 2 m. Thus, the Extreme scenario did not meet our actionable science standard at the time S17 was published.

The withdrawal of the Extreme scenario in S22 supports our argument for greater caution in presenting low-confidence science. Furthermore, our actionable science criterion suggests that the High scenario of 2.0 m by 2100 presented in S22 should not be regarded as actionable by practitioners, since it relies in a non-transparent way on low-confidence studies.

We would also like to reply to CW's statement that the Extreme scenario of 2.5 m was included "not because it is at all likely, but precisely because it is very, very unlikely". We do not object to presenting unlikely scenarios when there is a scientific basis for estimating likelihood. Rather, we object to presenting a single set of misleadingly precise probabilities, especially when these probabilities lack a physical underpinning and draw from low-confidence analyses.

Finally, while my main concern is helping the authors correct this particular error, I do also largely agree with the criticisms outlined in Community Comment 1 (CC1: '"Actionable" for whom, in what decision context?', Robert Kopp, 15 Mar 2024). It would be good to see the authors respond to and/or address those in their revision.

We have responded to Dr. Kopp's criticisms in a separate document.

I appreciate the authors spending the time and effort to grapple with these issues in the literature. I continue to be very supportive of having these types of issues and ideas discussed, and I believe the continuation of the dialogue through this paper is valuable.

We thank Dr. Weaver for joining us in grappling with these issues.

**References**

Bamber, J. L. and Aspinall, W. P.: An expert judgement assessment of future sea level rise from the ice sheets, Nature Climate Change, 3, 424–427, https://doi.org/10.1038/nclimate1778, 2013.

Behar, D., Kopp, R., DeConto, R., Weaver, C., White, K., May, K., and Bindschadler, R.: Consensus Statements: Planning for Sea Level Rise: An AGU Talk in the Form of a Co-Production Experiment Exploring Recent Science, https://www.wucaonline.org/assets/pdf/pubs-sfpuc-agu-consensus-statement.pdf, last access: 28 May 2024, 2017.

Boston Research Advisory Group: Climate Ready Boston: Climate Change and Sea Level Projections for Boston, https://www.boston.gov/sites/default/files/file/2023/03/2016_climate_ready_boston_report.pdf, last access: 28 May 2024, 2016.

Boyle, R., Hirschfeld, D., and Behar, D., Sea-Level Rise Practitioner Workshop Report: Leading Practices and Current Challenges. From Practitioner-led Workshops to Advance Resilience to Sea-Level Rise: Leading Practices and Current Challenges, https://doi.org/10.26077/npej-vw36, last access: 28 May 2024, 2022.

California Ocean Protection Council: State of California Sea-Level Rise Guidance: 2018 Update, https://www.opc.ca.gov/webmaster/ftp/ pdf/agenda_items/20180314/Item3_Exhibit-A_OPC_SLR_Guidance-rd3.pdf, last access: 28 May 2024, 2018.

DeConto, R. M. and Pollard, D.: Contribution of Antarctica to past and future sea-level rise, Nature, 531, 591–597, https://doi.org/10.1038/nature17145, 2016.

DeConto, R., Pollard, D., Alley, R., Velicogna, I., Gasson, E., Gomez, N., Sadai, S., Condron, A., Gilford, D., Ashe, E., Kopp, R., Li, D., and Dutton, A.: The Paris Climate Agreement and future sea-level rise from Antarctica, Nature, 593, 83–89, https://doi.org/10.1038/s41586- 021-03427-0, 2021.

Griggs, G., Árvai, J., Cayan, D., DeConto, R., Fox, J., Fricker, H. A., Kopp, R. E., Tebaldi, C., and Whiteman, E. A.: Rising Seas in California: An Update on Sea-Level Rise Science, https://digitalcommons.humboldt.edu/cgi/viewcontent.cgi?article=1005&context=hsuslri_state, last access: 28 May 2024, 2017.

Grinsted, A., S. Jevrejeva, R.E.M. Riva, and D. Dahl-Jensen, Sea level rise projections for northern Europe under RCP 8.5, Climate Research 64:15–23, https://doi.org/10.3354/cr01309, 2015.

Hirschfeld, D., Behar, D., Nicholls, R.J. et al. Global survey shows planners use widely varying sea-level rise projections for coastal adaptation. Communications Earth & Environment, 4, 102, https://doi.org/10.1038/s43247-023-00703-x, 2023.

Hirschfeld, D., Boyle, D, Nicholls, R.J., Behar, D., Esteban, M., Hinkel, J., Smith, G., and Hanslow, D.J.: Practitioner needs to adapt to Sea-Level Rise: Distilling information from global workshops, Climate Services, 34, 100452, https://doi.org/10.1016/j.cliser.2024.100452, 2024.

Jackson, L.P. and S. Jevrejeva, S., A probabilistic approach to 21st century regional sea-level projections using RCP and High-end scenarios. Global and Planetary Change, 146, 179–189, https://doi.org/10.1016/j.gloplacha.2016.10.006, 2016.

Jevrejeva, S., Grinsted, A., and Moore, J.C., Upper limit for sea level projections by 2100. Environ. Res. Lett., 9(10), 104008, https://doi.org/10.1088/1748-9326/9/10/104008, 2015.

Kopp, R. E., Horton, R.M, Little, C.M., Mitrovica, J.X., Oppenheimer, M., Rasmussen, D.J., Strauss, B., and Tebaldi, C.,Probabilistic 21st and 22nd century sea-level projections at a global network of tide-gauge sites, Earth's Future, 2(8), 383–406, https://doi:10.1002/2014EF000239, 2014.

Miller, K. G., Kopp, R.E., Horton, B.P., Browning, J. V., and Kemp, A.C., A geological perspective on sea-level rise and impacts along the U.S. mid-Atlantic coast, Earth's Future, 1, 3–18, https://doi.org/10.1002/2013EF000135, 2013.

Pfeffer, W. T., Harper, J. T., and Neel, S.: Kinematic constraints on glacier contributions to 21st-century sea-level rise, Science, 321, 1340–1343, https://doi.org/10.1126/science.1159099, 2008.

Rohling, E. J., Haigh, I.D., Foster, G.L., Roberts, A.P., and Grant, K.M., A geological perspective on potential future sea-level rise, Nat. Sci. Rep., 3, 3461, https://doi.org/10.1038/srep03461, 2013.

Sriver, R. L., N. M. Urban, R. Olson, and K. Keller, Toward a physically plausible upper bound of sea-level rise projections, Clim. Chang., 115, 893–902, https://doi.org/10.1007/s10584-012-0610-6, 2012.

Stammer, D., van de Wal, R.S.W., Nicholls, R.J., Church, J.A., Le Cozannet, G., Lowe, J.A., Horton, B.P., White, K., Behar, D., and Hinkel, J., Framework for high-end estimates of sea-level rise for stakeholder applications,  Earth's Future, 7, 923–938, https://doi.org/10.1029/2019EF001163, 2019.

Sweet, W. V., Kopp, R. E., Weaver, C. P., Obeysekera, J., Horton, R. M., Thieler, E. R., and Zervas, C.: Global and Regional Sea Level Rise Scenarios for the United States, Tech. Rep. NOS CO-OPS 83, National Oceanic and Atmospheric Administration, National Ocean Service, Silver Spring, MD, https://doi.org/10.7289/v5/tr-nos-coops-083, 2017.

Sweet, W. V., Hamlington, B. D., Kopp, R. E., Weaver, C. P., Barnard, P. L., Bekaert, D., Brooks, W., Craghan, M., Dusek, G., Frederikse, T., Garner, G., Genz, A. S., Krasting, J. P., Larour, E., Marcy, D., Marra, J. J., Obeysekera, J., Osler, M., Pendleton, M., Roman, D., Schmied, L., Veatch, W., White, K. D., and Zuzak, C.: Global and Regional Sea Level Rise Scenarios for the United States: Updated Mean Projections and Extreme Water Level Probabilities Along U.S. Coastlines, Tech. Rep. NOS 01, National Oceanic and Atmospheric Administration, National Ocean Service, Silver Spring, MD, https://oceanservice.noaa.gov/hazards/sealevelrise/noaa-nos-techrpt01-global-regional-SLR-scenarios-US.pdf, last access: 28 May 2024, 2022.

van de Wal, R. S. W., Nicholls, R. J., Behar, D., McInnes, K., Stammer, D., Lowe, J. A., Church, J. A., DeConto, R., Fettweis, X., Goelzer, H., Haasnoot, M., Haigh, I. D., Hinkel, J., Horton, B. P., James, T. S., Jenkins, A., LeCozannet, G., Levermann, A., Lipscomb, W. H., Marzeion, B., Pattyn, F., Payne, T., Pfeffer, T., Price, S. F., Seroussi, H., Sun, S., Veatch, W., and White, K.: A high-end estimate of sea-level rise for practitioners, Earth's Future, 10, e2022EF002 751, https://doi.org/10.1029/2022EF002751, 2022.

---

## Author Comment (AC2)

**Reply to Robert Kopp**

We thank Dr. Kopp for sharing his perspective on our manuscript. Our responses are below, with his comments in black font and our replies in blue.

\*\*\*\*\*

I read this brief comment with interest, and found some core issues troubling.

Fundamentally, the authors discuss 'actionable' science, but they discuss it stripped of context. Actions are defined by the American Heritage Dictionary as 'organized activity to accomplish an objective'. Science cannot be judged to be actionable, or not, outside the context of an organized activity and an objective. It makes little sense to talk about something being 'actionable' in general, outside of a specific decision context.

We agree in part. In the context of adaptation planning, we acknowledge that there are many financial, socioeconomic, and political factors driving decisions.

Nonetheless, we think it is important to distinguish between scientific claims that are sufficiently established to underpin decisions with significant fiscal, sociopolitical, and community implications, and those claims that are not. Low-confidence sea-level projections are likely to be superseded within a few years by very different numbers. The volatility of these projections (for instance, the excursion in high-end SLR projections from 2.0 m to 2.5 m and back within less than a decade, as shown in our Fig. 1) makes them risky to use for *any* decision where public confidence and large sums of money are at stake.

The authors neglect the extensive literature on decision science and risk analysis relevant to using sea-level projections in adaptation decision making. For a relatively recent review, see Keller, K., Helgeson, C., & Srikrishnan, V. (2021). Climate risk management. Annual Review of Earth and Planetary Sciences, 49, 95-116, https://www.annualreviews.org/doi/abs/10.1146/annurev-earth-080320-055847.

In the specific context of communicating sea-level uncertainty and ambiguity, the authors should also see Kopp, R. E., Oppenheimer, M., O'Reilly, J. L., Drijfhout, S. S., Edwards, T. L., Fox-Kemper, B., ... & Xiao, C. (2023). Communicating future sea-level rise uncertainty and ambiguity to assessment users. Nature climate change, 13(7), 648-660, https://www.nature.com/articles/s41558-023-01691-8. Given the direct relevance, this latter omission is particularly surprising.

The manuscript includes more citations than are typical for a Brief Communication, so we tried to be sparing in the number of works cited. With the Editor's approval, we will add these two citations to the revised manuscript.

Why do the organized activity and the objective matter?

Broadly, high-end sea-level rise scenarios, including low-confidence processes, are valuable in flexible, adaptive decision-making.

First, we would like to clarify a misunderstanding that may have arisen from our omissions in the submitted manuscript. We fully support flexible, adaptive decision-making, and we will make this clear in the revised manuscript.

We disagree about the value of low-confidence processes in decision-making. We are comfortable with including *low-likelihood* processes, if the likelihoods are scientifically supported (i.e., the relevant processes are understood with at least medium confidence). We object, however, to including processes which are so poorly understood that it is not yet possible to make robust, quantitative projections. Giving premature credence to low-confidence processes can lead to misuse of scarce public and private resources and can damage the credibility of the climate science enterprise.

This is shown by a number of papers, but perhaps most clearly and directly for this context in a preprint by Feng et al. (https://doi.org/10.22541/essoar.170914510.03388005/v1 ).

Among other analyses, Feng et al. compare idealized protection schemes for Manhattan under (1) a static optimal approach, where a single sea wall elevation must be picked based on available knowledge today, and (2) a variety of dynamic approaches, where sea wall height can be periodically adjusted based on new information. (I focus particularly on the 'reinforcement learning' approach described therein).

They consider two cases where projects are planned under inaccurate sea-level rise projections: (A) where planning takes place under the SSP5-8.5 low-confidence projections but the reality corresponds to SSP2-4.5 medium-confidence projections, and (B) where planning takes place under the SSP2-4.5 medium-confidence projections by reality corresponds to the SSP5-8.5 low-confidence projections.

In the former case -- where high-end projections are used and reality underperforms -- the expected net present value cost is $2.3 billion, $1.0 billion more than with the correct (lower) distribution, if a static approach is taken. With a flexible approach, the expected net present value cost is $1.0 billion, just $0.1 billion more than if the correct distribution is chosen.

However, in the latter case -- where middle-of-the-road projections are used and reality overperforms -- the expected net present value cost is $15 billion, $12 billion more than with the correct (high-end) distribution if a static optimal approach is taken. With a flexible approach, the expected net present value cost is $3.9 billion, $0.9 billion more than if the high-end distribution had been used. *[Here, RK included a table.]*

Thus, with a dynamic approach, using high-end projections that capture low-confidence processes makes a lot of economic sense. Such an approach cuts off the tail risk at relatively small additional cost. (In fact, the cost of a static optimal approach using the correct distribution in a middle-of-the-road world is more than the cost of using a dynamic approach with the overestimated, high-end distribution.)

However, with a static approach, the costs of getting the distribution wrong are more substantial (though an order of magnitude larger if the distribution is underestimated than if it is overestimated).

We agree that a flexible, dynamic approach is better than a static approach. We also think that capturing low-likelihood (as opposed to low-confidence) processes makes economic sense. When the science suggests the possibility of a low-likelihood, high-impact event, this should be included in planning. In the case of sea level, an optimal approach might be to start with an intermediate projection of (say) 1 m SLR by 2100, with the option to revisit this decision later based on new scientific understanding and relevant events (e.g., revised carbon emission pledges).

As discussed in the manuscript, the policies adopted by California in the wake of DeConto and Pollard (2016; hereafter DP16) and Sweet et al. (2017; hereafter S17) did not incorporate flexible, adaptive planning. Griggs et al. (2017) recommended use of the high-end 2.5 m scenario from S17, adjusted higher (e.g., 3.1 m for San Francisco) based on regional factors. The subsequent policy guidance in California OPC (2018) stated that practitioners should apply the high-end estimate to any assets whose failure "would have considerable public health, public safety, or environmental impacts". This guidance made no exceptions for projects with adaptive capacity (e.g., assets that could be relocated at moderate cost if the most pessimistic SLR projections are borne out).

So while we agree that a static approach can drive up costs unnecessarily, we would argue that current static policies in some jurisdictions, including California, are much more likely to err on the side of costly overbuilding rather than underbuilding, as a result of overreliance on low-confidence science.

In truth, I think the concern the authors address is not one with scientists offering practitioners low-confidence, high-end projections as part of the domain of plausible futures. It is with how these projections are then used.

This is true only in part. Indeed, it matters how the projections are used, but we are also concerned about how they are communicated by scientists. We have already discussed (above, and in our reply to Christopher Weaver) the undesired impacts from the embrace of DP16 in the Rising Seas report (Griggs et al., 2017, co-authored by RK). Our manuscript endorses the AR6 approach to low-confidence, high-end sea-level projections (Fox-Kemper et al., 2021). AR6

clearly distinguishes high-end projections based on medium-to-high-confidence science from those based on low-confidence science. This separation is valuable for long-term adaptation planning.

Further, for each projection, AR6 quantifies the contribution from each major source: thermal expansion, the Greenland and Antarctic ice sheets, mountain glaciers, and land water storage. (See, e.g., Table 9.9 in AR6.) This makes it fairly straightforward to adjust the global projections if subsequent evidence suggests a higher or lower contribution from a specific source. In contrast, the High and Extreme scenarios in S17 do not itemize the contribution from each source or the fraction of each contribution linked to low-confidence processes. This makes the global projections less transparent and thus harder for practitioners to use.

As the Feng et al. analysis, and others, indicate, the most economic approach given substantial uncertainty and ambiguity is most often the dynamic one.

We agree, and we will clarify this point in the revised manuscript.

Where a static approach must be used, whether due to inability to undertake a dynamic approach or regulatory inflexibility, then benefit-cost theory tells us what needs to be taken into account in order to determine the best option. This includes:

1) The benefit in terms of reduced risk associated with choosing different adaptation levels
2) The cost in terms of additional adaptive expenditures in terms of choosing different adaptation levels
3) The discount rate used to tradeoff present adaptation costs and future harms
4) The risk aversion that determines how much weight is given to the high-end of the cost distribution
5) The ambiguity aversion that determines how much weight is given to different alternative probability distributions for sea level and thus for cost.

We agree with these statements when the science is understood well enough to quantify benefits and costs. However, we disagree with the underlying assumption that high-end projections based on low-confidence processes are sufficiently constrained to support quantitative risk assessments. See, for example, the quotations from the Consensus Statements (Behar et al., 2017) in our reply to Christopher Weaver.

Where the costs and benefits of adaptation are comparable, discomfort will arise if regulatory guidance specifies a single adaptation target stripped of context, because a user's risk and ambiguity aversion applies to both the costs and benefits of adaptation, not just the benefits.

We agree with this statement. As stated above, we think the guidance adopted in California and some other jurisdictions is inappropriate, in part because it specifies a single target (or a narrow range of high targets) stripped of context.

I suspect that the authors' concern with the actionability of projections incorporating low-confidence processes is misaimed. Given appropriately flexible decision frameworks, as Feng et al. show, we are better off incorporating such high-end projections.

We agree that flexible decision frameworks make it possible to live with uncertainty and adapt to advancing knowledge. Also, as our manuscript states, we have no objection to incorporating high-end projections, if supported by medium- or high-confidence science as evaluated in an appropriate community-based process.

We think our basic disagreement with RK is in how to decide which science is sufficiently accepted to underpin adaptation planning. We are trying to set forth a standard that is philosophically justified and practical to implement.

In his critique, RK implies that our standard is too strict, because it rules out science that (in his view) should be included in planning. However, he does not articulate a clear alternative.

We think some standard is necessary. Otherwise, any claims—no matter how outlandish—could be passed on to practitioners. We are not clear on the standard used by S17, who cited nine studies in support of the Extreme scenario. (Our response to Christopher Weaver explains why we think these studies were not actionable.) Each study appeared in a reputable peer-reviewed journal, which perhaps suggests an alternative standard.

However, we do not think this is the standard actually used by S17, because S17 omitted some relevant peer-reviewed science. Consider, for example, the study of Hansen et al. (2016), which received wide press coverage and was probably known to the S17 authors. This study claimed that mass loss from ice sheets "is better approximated as exponential than by a more linear response". The authors suggested that a doubling time of 10 years is plausible, yielding 5 m of global mean SLR by 2100 (see their Fig. 5).

Had S17 cited this study, they could have argued for a high-end scenario much greater than 2.5 m. Why did they exclude it? We cannot say for sure, but we suspect that they did not find the study credible, perhaps because of criticism from other climate scientists (as highlighted by Revkin, 2015). Thus, they did not deem it suitable for planning in decision contexts. This suggests that S17 had an evidential standard, albeit with a lower threshold than ours. If so, then we disagree not on whether a threshold of evidence should exist, but on where it should be set.

We would like to quote from Rajashree Datta's comments:

> Presumably, we would not present decision-makers with non-peer-reviewed SLR estimates and expect them to decide its merit based on the specific decision context. If we accept the current social production of science which is "peer review" (also imperfect), a higher standard for "actionable science" is simply a logical extension…. In another comment, Dr. Kopp suggests (in summary) that it is critical to present the long tail and leave room for a dynamic response, even where evidence is lacking, based on the extent of potential risk (and the associated benefit of more extensive adaptation). I see no meaningful contradiction between the need for a guideline presented by the authors and the presence of exceptions. In fact, the "exceptionality" here is still defined in reference to some guideline and underlying rationale, thus underlining the need for the guideline.

We agree that while it might not be possible to state guidelines that should apply without exception in all cases, evidence-based guidelines are still needed. We think that as a general rule, low-confidence science is not an adequate foundation for adaptation planning. The burden of argument then falls on those who think a low-confidence standard is justified in a specific context.

Both Stammer et al. (2019) and van de Wal et al. (2022) took on this question, with results that reinforce the value of separating medium- and high-confidence projections from low-confidence projections. Our effort seeks to amplify and deepen these perspectives. We suggest that those who seek different standards can propose their own criteria.

Regulations that rigidly prescribe the use of specific high-end projections in static contexts, however, run the risk of leading to sub-optimal outcomes.

We agree, and we are concerned that such regulations are still in place in the jurisdictions mentioned.

It may be appropriate for policy to set discount, risk aversion, and ambiguity aversion levels for specific contexts; this is a matter where different political philosophies will lead to different judgements. However, given these parameters, identifying the benefit-cost optimal outcome requires considering the net value of adaptation benefits and adaptation costs under these parameters. If costs and benefits are comparable, overly rigid targets might cut off the long tail of sea-level harms but create a long tail of adaptation cost overruns.

In short, the authors have chosen the wrong target. Scientists should strive to communicate not just projections that incorporate processes for which there is a high degree of evidence, but also processes that are of potentially great significance but less agreement and evidence -- as AR6 has done. It is, however, important that actions be guided by decision frameworks that correctly reflect the nature of the information provided.

We agree that there is value in communicating low-confidence projections, provided the contributions of low-confidence processes are clearly separated from those based on medium- and high-confidence processes. That is, we favor the approach of AR6 over that of S17 and its followup report, Sweet et al. (2022). We agree that practitioners should be aware of the current state of the science, including cutting-edge research like DP16. But in nearly all cases, we would discourage practitioners from using low-confidence projections for decision-making because of the risks discussed in the manuscript: confusion, maladaptation, and whiplash leading to loss of public confidence.

We hope these responses will clarify several points on which we agree with Dr. Kopp, along with some points of real disagreement.

**References**

Behar, D., Kopp, R., DeConto, R., Weaver, C., White, K., May, K., and Bindschadler, R.: Consensus Statements: Planning for Sea Level Rise: An AGU Talk in the Form of a Co-Production Experiment Exploring Recent Science, https://www.wucaonline.org/assets/pdf/pubs-sfpuc-agu-consensus-statement.pdf, last access: 28 May 2024, 2017.

California Ocean Protection Council: State of California Sea-Level Rise Guidance: 2018 Update, https://www.opc.ca.gov/webmaster/ftp/pdf/agenda_items/20180314/Item3_Exhibit-A_OPC_SLR_Guidance-rd3.pdf, last access: 28 May 2024, 2018.

Fox-Kemper, B., Hewitt, H. T., Xiao, C., Aðalgeirsdóttir, G., Drijfhout, S. S., Edwards, T. L., Golledge, N. R., Hemer, M., Kopp, R. E., Krinner, G., Mix, A., Notz, D., Nowicki, S., Nurhati, I. S., Ruiz, L., Sallée, J.-B., Slangen, A. B. A., and Yu, Y.: Ocean, Cryosphere and Sea Level Change, in: Climate Change 2021: The Physical Science Basis. Contribution of Working Group I to the Sixth Assessment Report of the Intergovernmental Panel on Climate Change, edited by Masson-Delmotte, V., Zhai, P., Pirani, A., Connors, S., Péan, C., Berger, S., Caud, N., Chen, Y., Goldfarb, L., Gomis, M., Huang, M., Leitzell, K., Lonnoy, E., Matthews, J., Maycock, T., Waterfield, T., Yelekçi, O., Yu, R., and Zhou, B., p. 1211–1362, Cambridge University Press, Cambridge, United Kingdom and New York, NY, USA, https://doi.org/doi:10.1017/9781009157896.011, 2021.

Griggs, G., Árvai, J., Cayan, D., DeConto, R., Fox, J., Fricker, H. A., Kopp, R. E., Tebaldi, C., and Whiteman, E. A.: Rising Seas in California: An Update on Sea-Level Rise Science, https://digitalcommons.humboldt.edu/cgi/viewcontent.cgi?article=1005&context=hsuslri_state, last access: 28 May 2024, 2017.

Hansen, J., Sato, M., Hearty, P., Ruedy, R., Kelley, M., Masson-Delmotte, V., Russell, G., Tselioudis, G., Cao, J., Rignot, E., Velicogna, I., Tormey, B., Donovan, B., Kandiano, E., von Schuckmann, K., Kharecha, P., Legrande, A. N., Bauer, M., and Lo, K.-W.: Ice melt, sea level rise and superstorms: evidence from paleoclimate data, climate modeling, and modern observations that 2 °C global warming could be dangerous, Atmos. Chem. Phys., 16, 3761–3812, https://doi.org/10.5194/acp-16-3761-2016, 2016.

Revkin, A.: A Rocky First Review for a Climate Paper Warning of a Stormy Coastal Crisis, New York Times, https://archive.nytimes.com/dotearth.blogs.nytimes.com/2015/07/25/a-rocky-first-review-for-a-climate-paper-warning-of-a-stormy-coastal-crisis/?_r=0, last access: 28 May 2024, 2015.

Stammer, D., van de Wal, R.S.W., Nicholls, R.J., Church, J.A., Le Cozannet, G., Lowe, J.A., Horton, B.P., White, K., Behar, D., and Hinkel, J., Framework for high-end estimates of sea-level rise for stakeholder applications, Earth's Future, 7, 923–938, https://doi.org/10.1029/2019EF001163, 2019.

Sweet, W. V., Kopp, R. E., Weaver, C. P., Obeysekera, J., Horton, R. M., Thieler, E. R., and Zervas, C.: Global and Regional Sea Level Rise Scenarios for the United States, Tech. Rep. NOS CO-OPS 83, National Oceanic and Atmospheric Administration, National Ocean Service, Silver Spring, MD, https://doi.org/10.7289/v5/tr-nos-coops-083, 2017.

Sweet, W. V., Hamlington, B. D., Kopp, R. E., Weaver, C. P., Barnard, P. L., Bekaert, D., Brooks, W., Craghan, M., Dusek, G., Frederikse, T., Garner, G., Genz, A. S., Krasting, J. P., Larour, E., Marcy, D., Marra, J. J., Obeysekera, J., Osler, M., Pendleton, M., Roman, D., Schmied, L., Veatch, W., White, K. D., and Zuzak, C.: Global and Regional Sea Level Rise Scenarios for the United States: Updated Mean Projections and Extreme Water Level Probabilities Along U.S. Coastlines, Tech. Rep. NOS 01, National Oceanic and Atmospheric Administration, National Ocean Service, Silver Spring, MD, https://oceanservice.noaa.gov/hazards/sealevelrise/noaa-nos-techrpt01-global-regional-SLR-scenarios-US.pdf, last access: 28 May 2024, 2022.

van de Wal, R. S. W., Nicholls, R. J., Behar, D., McInnes, K., Stammer, D., Lowe, J. A., Church, J. A., DeConto, R., Fettweis, X., Goelzer, H., Haasnoot, M., Haigh, I. D., Hinkel, J., Horton, B. P., James, T. S., Jenkins, A., LeCozannet, G., Levermann, A., Lipscomb, W. H., Marzeion, B., Pattyn, F., Payne, T., Pfeffer, T., Price, S. F., Seroussi, H., Sun, S., Veatch, W., and White, K.: A high-end estimate of sea-level rise for practitioners, Earth's Future, 10, e2022EF002 751, https://doi.org/10.1029/2022EF002751, 2022.

---

## Author Comment (AC3)

**Responses to Reviewers 2 and 3**

We thank the reviewers for their thoughtful and constructive comments. Here, we will respond to the comments from Reviewer 2 (Rebecca Priestley) and Reviewer 3. We responded previously to Reviewer 1 (Chris Weaver).

A few introductory remarks: We have prepared a revised version of the manuscript with several significant changes in response to the reviews and community comments:

- In the Introduction, we added a discussion of decision-making under deep uncertainty (DMDU), and the use of low-confidence science within DMDU. We have clarified that when we say a scientific claim is actionable, we mean that it justifies adaptation actions (i.e., physical measures and financial investments) in the near term. If a claim is not actionable based on current evidence, it can still be used fruitfully to explore future options in DMDU frameworks, but it should be differentiated from claims backed by more robust evidence.
- Accordingly, we reworded the criterion for actionable science in Section 2. The new criterion reads, "*A scientific claim is sufficiently accepted to justify adaptation action (i.e., near-term physical measures and financial investments) when it is supported by multiple, consistent independent lines of high-quality evidence leading to high or medium confidence, as determined by a diverse group of experts in an open, transparent process.*"
- Based on Chris Weaver's comments, we clarified that the source of the 2.5 m Extreme Scenario in Sweet et al. (2017) was the probabilistic projection in Kopp et al. (2014), using the Antarctic projections of Bamber & Aspinall (2013). The modified text describes a narrower role for DeConto & Pollard (2016) in S17.
- We divided the previous Section 3 into two sections. The new Section 3 focuses on scientific projections (mainly DP16), and the new Section 4 discusses the communication of these projections to practitioners.
- The revised Section 4 includes some guidance on how practitioners might work with low-confidence sea-level projections in the future, given that there is not always a clear distinction between near-term adaptation actions and long-term contingency plans.

The rest of this document includes our responses to Reviewers 2 and 3. Responses are in blue font.

**Reviewer 2 (Rebecca Priestley):**

I found this paper very interesting. I am familiar with DeConto and Pollard's 2016 paper, and the subsequent media coverage, but was not aware of the extent to which these projections were taken on by policymakers and practitioners. This case study aspect of the paper is very interesting and valuable (though I note the corrections advised by Chris P. Weaver in the

interactive review). I also found the comments about disciplinary journals vs high impact journals (lines 54-64) particularly valuable.

This has potential to be an important paper, so my feedback is quite detailed with much of it focused on precision of language, to ensure clear and purposeful communication of the argument of the paper.

We appreciate the reviewer's attention to precise language.

*Specific comments on section 2 of the paper*

I have specific comments about language use in section 2 of the paper, firstly around use of the word hypothesis (eg, line 68: 'transform novel hypotheses into accepted knowledge', line 99: 'A scientific hypothesis is sufficiently accepted for use in decision-making when it is supported by ... etc' and line 102: 'peer-reviewed hypotheses must be scrutinized by a diverse group of scientists etc'). I was surprised by the focus on the word 'hypothesis' here. Not all science starts with a hypothesis, and even when it does, this word is usually used to describe what comes at the start of a study, not the end. I would have thought that it's not the 'peer-reviewed hypothesis' that is the 'actionable' (or not) part of a research project, or the resulting paper. Rather, it's the peer reviewed conclusions, claims, findings, or theories. Or, as the quote from Behar says, the 'data, analysis and forecasts' (line 25).

We agree with the reviewer. In several instances, including ll. 24, 68, 99, 102, 204 and footnote 2 of the original manuscript, we changed "hypotheses" to "claims". We retained one instance of "hypotheses" when discussing Longino's work, since she herself uses that word. We think "claims" is more exact than "theories" (which has a broader meaning than what we want to convey) or "findings" (which does not as clearly connote the presence of uncertainty).

In the same section, I also suggest a review of the words 'viewpoints', 'opinion', and 'assumptions'. Scientists do, of course, have viewpoints, opinions, and assumptions, but this paper is focused on peer-reviewed published research which (we hope!) relies on evidence and observations that lead to claims and conclusions (even if it doesn't meet the criteria for actionable research). At the moment the paper could imply that scientists make claims in their published research, or IPCC authors make decisions, based only on opinions and assumptions (which could feed into politically motivated narratives seeking to undermine climate science).

I realise that different disciplines have different norms about language use, but with an interdisciplinary paper like this it's important that the meaning of this language is accessible to a broad readership. I suggest therefore that language use is reviewed, especially around the words I've mentioned here.

We thank the reviewer for this suggestion. We removed "viewpoints" and "opinions" on l. 74. We kept "assumptions" on ll. 47 and 76, since the process of critical scientific review to challenge implicit background assumptions is central to Longino's analysis.

*Specific comments on section 3 of the paper*

The first paragraph of section 3 is important, but is not communicating as clearly as it could. In lines 108-110 I suggest removing reference to 'land ice'. At the moment, the AIS is listed as an example of 'land ice' in one sentence, then the next sentence says it 'contains marine-grounded ice'. To avoid confusion, but not take away any meaning, the reference to 'land ice' could be removed and the more standard separation of SLR contributors into thermal expansion, mountain glaciers, the Greenland Ice Sheet and AIS used (as has been done in line 180). Then, in line 110, which says 'if melted, this ice could raise sea level by several meters', it needs to be explicit what ice is being referred to here.

We agree, and we revised the paragraph as follows:

> Global mean sea level (GMSL) is rising by about 3.7 mm/yr, mainly because of ocean thermal expansion and the loss of ice from the Greenland and Antarctic ice sheets (GrIS and AIS) and mountain glaciers (Fox-Kemper et al., 2021). Uncertainty in long-term sea-level projections is dominated by the AIS, which contains a large volume of ice that is grounded below sea level and is vulnerable to retreat under climate warming. If melted, this Antarctic ice could raise sea level by several meters.

In line 140 it would be useful to provide the figures for the De Conto 2021 lowered 21st century SLR contribution, to allow comparison with the DP16 figure.

Thanks for the suggestion. We revised the text to read, "In a follow-up to DP16, DeConto et al. (2021) revised the atmospheric forcing, delaying hydrofracture and lowering the projected 21$^{st}$ century AIS sea-level contribution to 0.5 m or less, even if MICI is active." This figure comes from Table 1 of that study, which gives a median Antarctic contribution of 0.34 m and a range of 0.20–0.53 m under RCP8.5. Thus, the high-end Antarctic contribution in DeConto et al. (2021) is reduced by about 0.5 m compared to DP16.

Line 71: states that IPCC assessments 'are directed mainly to policymakers but are read by practitioners' – I suggest that the difference between policymakers and practitioners is teased out in this paper, and more emphasis given to the role of policymakers. The publications referred to in section 3 seem to be interpretations by policymakers, that were then actioned by practitioners. In other parts of the paper, though, the emphasis on practitioners suggests that they are actioning science without this layer of interpretation by policymakers. For example in line 213 is it primarily practitioners or policymakers who need to 'view novel peer-reviewed claims with caution'? In line 225, is it 'scientists and practitioners' who need to work more closely together, or scientists and policymakers?

We regret the confusion. We think of policymakers as the people who make laws and regulations, such as limits on greenhouse gas emissions. In general, policymakers would have more official power than practitioners and would be less involved in on-the-ground planning and implementation. (A similar distinction can be made between "decision-makers" and practitioners.) Both policymakers and practitioners lie on the receiving end of scientific communication; policymakers generally would not serve as intermediaries between scientists

and practitioners. The publications in Section 3 were written mainly by scientists from universities and government labs, with some representatives from the practitioner community.

We added the following text in footnote 1: "We think of practitioners as distinct from policymakers: the legislators and other government officials who make laws and regulations."

On l. 213, it is primarily practitioners who should view novel peer-reviewed claims with caution. (We would encourage policymakers to exercise caution also, but they are not our main audience.) On l. 225, we think that scientists and practitioners should work more closely together.

*Technical corrections and points of clarification*

Line 27: says the term 'actionable science' (which I was not familiar with) has been 'widely adopted' but there's only one citation here. More citations here would strengthen this claim.

We added the following citations, which refer to "actionable climate science" and "actionable climate information", respectively:

Executive Office of the President, 2013. The President's Climate Action Plan. Available: https://obamawhitehouse.archives.gov/sites/default/files/image/president27sclimateactionplan.pdf

WCRP Joint Scientific Committee (JSC), 2019. World Climate Research Programme Strategic Plan 2019–2028. WCRP Publication 1/2019. Available: https://www.wcrp-climate.org/images/documents/WCRP_Strategic_Plan_2019/WCRP-Strategic-Plan-2019-2028-FINAL-c.pdf

Others are available, including the USGCRP 2012-21 Strategic Plan, two Presidential Executive Orders, and a recent Biden-Harris administration press release regarding the Fifth National Climate Assessment, but we are mindful that the article already includes more citations than are standard for a Brief Communication.

Line 30: 'Our goal is to offer guidance ...' who to? Is this guidance for scientists, practitioners, or both?

We have clarified that the guidance is for both scientists and practitioners.

Line 38: As a science historian I have to note that the discipline is decades on from the 'lone genius' approach, as is much popular science history. This is perhaps a traditional approach, or a twentieth century approach, but I'm not sure it's right to say 'often' when referring to current work.

Our sense is that while historians and philosophers of science have moved on from this approach, popular accounts in climate science and other fields have been slower to catch up. But we agree there has been progress, so we changed "often" to "sometimes".

Lines 66, 67: Mentions first 'press releases' and then 'media accounts'. It would be good to explicitly make the connection between the press releases and the media accounts – while the press releases might cast the work in dramatic light, the media stories often go further, and the headlines (which are not written by the journalists) even further than that, with attention seeking headlines.

This is an excellent suggestion. Citing the study by Perga et al. (2023), we rewrote the first two sentences of this paragraph as follows:

> To the extent practitioners learn about climate research through media reports, their attention will likely be drawn to a small number of studies in high-impact journals, focused on 21$^{st}$ century global-scale threats (Perga et al., 2023). Press releases from journals and universities often cast the work in a dramatic light, and media stories with attention-seeking headlines heighten the drama.

We took the liberty of using "attention-seeking headlines" from your suggestion.

Do you have a citation for the statement that 'practitioners typically learn of scientific advances through media coverage'? (line 65)

DB has the experience of frequently receiving inquiries from fellow practitioners about media reports on new studies; these practitioners want to know whether the new studies change our basic understanding of SLR. However, we were not able to find a study that focuses on practitioner information-gathering, so we replaced this sentence with the wording above.

Line 90: citation and page number needed for this quote

The citation is Mastrandrea et al. (2010), the guidance note cited in the previous paragraph. We changed "this guidance" to "this guidance note" to make the reference more clear. In general, we have given page numbers when quoting from books (i.e., Longino) but not articles, but we can add page numbers (this one is on p. 2) if the editor thinks it would be helpful for readers.

Line 95: Makes an important point, but is it also worth noting that opting for 'higher ground' is not necessarily guarding against 'unknown risks', it could alternatively (or also) be seen as choosing an option with a longer lifespan, given that sea level rise will continue beyond 2100.

We agree that higher ground could extend the lifespan, but this perhaps is a distraction from the main point. We have rewritten the example to describe a levee with an expected lifetime of 75 years (i.e., until 2100). In this case, a decision to expand the levee footprint to potentially accommodate the greater SLR projected by a low-confidence study would not be justified by our criterion, unless the expansion was inexpensive.

Line 186: what does 'community' mean in this context? The scientific community?

Yes, we changed this to "scientific community".

Line 213: Is 'contradict' the right word here, or would 'challenge' be more appropriate?

Thanks, we changed this to "challenge".

I look forward to your response. As I said at the start, this is a very interesting paper.

Thank you again for your helpful comments.

**Reviewer 3:**

In this manuscript, Lipscomb et al. discuss the challenges of providing 'actionable' scientific research in the context of climate adaptation. In the manuscript, the authors emphasize the importance of distinguishing between novel hypotheses/claims and actionable science that can be used for decision-making. The authors discuss this (also) within the context of a recent high-impact study projecting rapid sea level rise from the Antarctic ice sheet due to a low-confidence process. Overall, Lipscomb et al. propose (1) an epistemic criterion for determining when scientific claims are actionable, based on multiple lines of high-quality evidence and evaluated by a diverse group of experts, (2) recommendations for scientists and practitioners to improve the use of actionable science in decision-making.

This manuscript has clearly attracted lots of interest and sparked productive discussions (see comment section and follow-up AGU presentation highlighted by Chris Weaver, CC2). The authors replied already quite extensively to the main comments raised by Robert Kopp (CC1) and Chris Weaver (CC2). If the authors are willing to include their reply to CC1 and CC2 in the revised version of the manuscript (in particular, the misstatement about the relevance of DP16 on the Sweet et al. (2017) report), I will consider it ready for publication as is; the manuscript is in general very well written and clearly an excellent fit for The Cryosphere (Brief Communication).

Yes, the revised manuscript responds to these comments and states that Kopp et al. (2014) was the source of the Extreme (2.5 m) scenario in Sweet et al. (2017). We stand by our statement that the citations of DP16 in S17 served to bolster support for this scenario, even though K14 was the original source.

I do have a couple of minor comments to add, which the authors can see as a suggestion for the revised manuscript. I do not consider these (minor) comments as strictly required for publication - I rather hope they can contribute to the discussion.

1) In general, I agree with the authors on how the epistemic criterion for actionable science is formulated, and with the recommendations to scientists, journalists, and practitioners laid out in Section 4. Both the criterion and the recommendations largely rely on IPCC reports or meta analyses/community assessment. While this makes sense to me, I'm skeptical about how the criterion and recommendations could be applied in practice, as IPCC reports (or other community assessments) are published at much longer timescales than individual studies, and

media are typically very quick to pick up high-impact claims (often at the same time a new study comes out for high-impact journals). Even assuming improved awareness and communication between scientists, journalists, and practitioners in the future (which is one recommendation made by the authors and is certainly something we should aspire to), it looks to me that for a case similar to the one presented by the authors to not happen again much would be left to individual choices (for instance: being cautious when making/dealing with new claims). I am fine if the main goal of the authors is to start a discussion on the topics presented, rather than proposing some examples of practical solutions to implement their recommendations. However, I think it would be of great help to see some (more) critical reflection on the latter. For instance: should it become part of the peer-review process to have reviewers providing some level of confidence and/or rating how much a study can be considered reliable or even suitable for media coverage (using for example a formal rating system similar to the one used to evaluate originality, quality, etc.)?

This is a good point, which the three of us have talked about. The idea of adding a rating of confidence or reliability rating is intriguing, but we decided not to recommend changes in the peer-review process. In part, this is because we are uncertain that reviewers are well positioned to assess confidence, given that confidence can arise from the convergence of multiple research findings across disciplines, not all of which a single reviewer would necessarily be familiar with. We think confidence is better assessed by multidisciplinary groups as in the IPCC process.

We hope that by reading our paper, scientists and practitioners will become more aware of publishing and media incentives that might result in misinterpretation of scientific claims during adaptation planning. We think that discussions between scientists and practitioners are an important step toward finding lasting solutions. Finally, our recommendation that projections deemed by IPCC to be of low confidence are not actionable provides criteria we believe practitioners, climate service providers, and researchers alike can apply.

2) Line 65: 'practitioners typically learn of scientific advances through media coverage'. I think this is quite an important point of focus in the manuscript - the link between scientific results/media coverage/practitioners. I think however that this sentence is a bit too vague, and it would be good to have reference(s) backing it up. If there aren't, maybe it could make sense to make the example for the DP16 study, but to avoid generalizing ( 'typically learn').

We did not find a direct citation for this claim, so we reworded the sentence as shown above, with a citation of Perga et al. (2023).

Thank you for sharing your suggestions on our manuscript.

---

## Author Comment (AC4)

**Responses to Community Comments**

We thank Robert Kopp, Jeremy Bassis, Judy Lawrence, Marjolijn Haasnoot, and Robert Lempert for their thoughtful and detailed comments. Our replies are below (in blue font). Line references are to the revised manuscript.

**Reply to Robert Kopp**

In my mind, the core of the authors' response is the following bold statement:

> "We disagree about the value of low-confidence processes in decision-making. We are comfortable with including low-likelihood processes, if the likelihoods are scientifically supported (i.e., the relevant processes are understood with at least medium confidence). We object, however, to including processes which are so poorly understood that it is not yet possible to make robust, quantitative projections. Giving premature credence to low-confidence processes can lead to misuse of scarce public and private resources and can damage the credibility of the climate science enterprise."

These are bold statements that appear to reject the entire field of decision making under deep uncertainty (DMDU). Scholarly practice would suggest that such a bold claim be made head on – that is, if the authors are to dismiss the entire DMDU literature, they need to cite and engage with that literature in a manner that justifies rejecting decades of scholarly work (and of practice), a relatively recent synthesis of which is provided by

> Marchau, V. A., Walker, W. E., Bloemen, P. J., & Popper, S. W. (2019). Decision making under deep uncertainty: from theory to practice. Springer Nature, 405 pp. https://doi.org/10.1007/978-3-030-05252-2

Contrary to this literature, the authors argue that the only appropriate decision-making practice regarding those elements of sea-level projection characterized by deep uncertainty (or low confidence) is to ignore them. I strongly disagree with this position, but would welcome an argument that makes it while engaging with this robust body of literature.

We regret the appearance that we are rejecting the large body of work on DMDU, including the excellent synthesis of Marchau et al., (2019). The revised manuscript describes DMDU and discusses its value. The Introduction now includes a brief overview of DMDU (ll. 40–47):

> Researchers and practitioners have developed frameworks for decision making under deep uncertainty (DMDU; Marchau et al., 2019a). Deep uncertainty arises when experts are unable to "specify the appropriate models to describe interactions among the system's variables, select the probability distributions to represent uncertainty about key parameters in the models, and/or value the desirability of alternative outcomes" (Lempert et al., 2003; Marchau et al., 2019b). DMDU frameworks support a shift from a "predict then act" paradigm to a "monitor and adapt" paradigm (Marchau et al., 2019b). In the

new paradigm, the focus is on exploring a wide range of plausible futures and committing to short-term actions, while keeping open long-term options that might be triggered by new evidence. Ideally, costly actions are deferred until they are necessary.

We are comfortable with the use of low-confidence science in a monitor-and-adapt paradigm, provided it is not prematurely held to be actionable. The revised Introduction also clarifies what we mean by "actionable" (ll. 33–38):

Here, we will say that a claim is actionable when it is sufficiently accepted to justify adaptation action in the near term (assuming that other requirements for actionability, such as salience and legitimacy, have also been met). Near-term actions—for example, physical measures such as building seawalls and levees, as well as financial investments such as acquiring land—may be needed not only to address short-term vulnerability but also to prepare for long-term climate impacts. Thus, uncertainties about the rate of climate change in the next several decades (to 2100 and sometimes beyond) must be factored into near-term decisions.

This language clarifies that we do not think low-confidence science should be ignored. It can be useful for dynamic adaptive planning even when it is not sufficient to justify near-term action.

The authors may also find it helpful to engage with the new manuscript: Lempert, R., Lawrence, J., Kopp, R., Haasnoot, M., Reisinger, A., Grubb, M., & Pasqualino, R. The Use of Decision Making Under Deep Uncertainty in the IPCC. Frontiers in Climate, 6, 1380054, https://doi.org/10.3389/fclim.2024.1380054).

Thank you for pointing out this paper, which is cited in the revised manuscript (ll. 241–243):

Lempert et al. (2024) noted that AR5 and AR6 "opened the aperture" to provide a wider range of possible futures by discussing low-confidence science in detail. Consistent with van de Wal et al. (2022), we argue that practitioners should embrace widening uncertainty in designing adaptation action only when justified by scientific confidence.

In other words, we suggest that the aperture be opened cautiously. If low-confidence claims are included in the range of possible futures, then the level of confidence should be communicated clearly to practitioners.

It is certainly the case that there has been – and often remains – broad disagreement among experts regarding technological and policy development processes that would justify applying the label of "low confidence." As Moss et al. (2010, https://doi.org/10.1038/nature08823) noted in laying out the RCP/SSP framework, "An underlying key issue [and, thus, point of low agreement and therefore low confidence] is whether probabilities can be usefully associated with scenarios or different levels of radiative forcing; for example, the probability that concentrations will stabilize above or below a specified level."

Do the authors think that the issue highlighted by Moss et al. must be answered affirmatively --
that is, we must show that probabilities can be usefully associated with emissions scenarios --
for climate projections driven by those scenarios to be actionable? If experts cannot agree on
probability distributions regarding policy and technological development, does that moot the
actionability of any climate projections -- which is to say, any climate projections beyond about a
20-30 year time horizon – that exhibit substantially sensitivity to these deeply uncertain
processes? If so, that somewhat moots the discussion of low-confidence ice-sheet processes,
which we have little reason to think might manifest at significant levels until late in the century.

No, we do not think that probabilities must be assigned to emissions scenarios in order for the
climate projections driven by these scenarios to be actionable. More generally, the lack of a pdf
does not imply low confidence. According to Mastrandrea et al. (2011), "In most cases, the
author team should have high or very high confidence in a finding characterized
probabilistically." We think that medium-confidence projections can be actionable, although in
many cases they cannot be characterized probabilistically. This is why our actionable science
criterion ("*multiple, consistent independent lines of high-quality evidence leading to high or
medium confidence, as determined by a diverse group of experts in an open, transparent
process")* does not mention probability.

We would not describe the CMIP emissions scenarios as having low confidence or being
insufficiently vetted by the community.  The ScenarioMIP activity leading to the CMIP6 scenarios
(O'Neill et al., 2016) is a good example of a diverse group of experts engaging in an open,
transparent process. Although high-end scenarios such as SSP5-8.5 are not described
probabilistically, they have been judged to be plausible based on a great deal of physical and
socioeconomic evidence. The scenarios themselves and the process producing the scenarios
are updated based on experience and new evidence (O'Neill et al., 2020). This is very different
from the rapid adoption of high-end sea-level projections described in our article.

I continue to believe that the authors' actual objection is not to the use of deeply uncertain
information in decision making, but the use of such information with decision frameworks that
are designed for probabilistic information.

It's true that we don't object to the use of deeply uncertain information in decision making, but
only to inappropriate use. The revised manuscript clarifies how we think this information can be
used appropriately.

**Reply to Jeremy Bassis**

This short paper describes a framework for actionable science.  I am writing this comment as a
glaciologist who also works broadly in the field of adaptation and science usability. I would really
like to encourage the authors and editors to expand the paper to full length to give service to the
ideas because in the short format currently there is much ambiguity and with significant room for
mischief and harm.

We think the Brief Communication format is appropriate for our arguments and intended audience. We regret that there was ambiguity in the initial submission as a result of some omissions and lack of clarity. We hope the revised manuscript will resolve the ambiguity while staying within the short format.

To start, the study of how knowledge systems are applied to decision making often goes under the name "science usability", but I have also seen it referred to as "actionable science".  There is substantial literature about the process of knowledge creation for planning and decision making. Foundational to this field, is the concept that for knowledge to be applied in a decision making or planning concept requires credibility, salience and legitimacy (see, e.g., Cash et al., 2003).  To quote directly from Cash et al., (2003) ". . . . *credibility* involves the scientific adequacy of the technical evidence and arguments. *Salience* deals with the relevance of the assessment to the needs of decision makers. *Legitimacy* reflects the perception that the production of information and technology has been respectful of stakeholders' divergent values and beliefs, unbiased in its conduct, and fair in its treatment of opposing views and interests."

The definition of "actionable" that the authors present only includes what would be traditionally called "credibility", omitting salience and legitimacy completely.  This is a fatal omission from a usability standpoint because there is abundant, credible literature that demonstrates the importance of salience and legitimacy (e.g., Cash, 2003). To give two examples of salience, communities adapt to local sea level not global sea level and thus global sea level rise might have low salience and thus low usability for many decision makers.  Or century and longer sea level rise projections have little salience for, say 30 year home mortgages. Legitimacy is also important as it impacts the process and values of science creation and there are numerous examples where low legitimacy jeopardized science usability.

Clearly usability depends on the specific decisions and the community making decisions and, as noted by another commenter,  usability cannot be divorces from this context: actionable in one decision context does not mean actionable in a different decision context. We also know that the way to increase the usability of knowledge is through co-generation, but that this is inefficient.  Usually it is knowledge brokers and boundary organizations that play the role of interpreters and translators of knowledge to users.  There is, again, abundant literature in this field and it is a shame to have no dialogue with the many previous studies that identify criteria AND the process of usable knowledge production.

My view is that the present study has *much* more value if this study is reframed more specifically around credibility rather than more broadly about usability/actionable. I would even recommend switching out the term "actionable" for "credible" to better reflect existing terminology.

These comments are based on a misunderstanding. It was never our intention to argue that credibility (or acceptance) is a sufficient condition for actionability, only that it is a necessary condition. Other writers, including Cash, have addressed the importance of salience and

legitimacy. We chose to focus on credibility/acceptance because there has been confusion among practitioners about what counts as "sufficiently accepted".

The revised manuscript includes the following revised text in the Introduction (ll. 29–34, with new text in italics):

> The WUCA definition hints at the kind of knowledge appropriate for driving adaptation action ("sufficiently accepted"), but does not guide practitioners in identifying this knowledge. *Similarly, Cash et al. (2003) argued that scientific information can motivate action to the extent it is seen as credible, salient, and legitimate, but they did not offer criteria for credibility.*
>
> *Here, we will say that a claim is actionable when it is sufficiently accepted to justify adaptation action in the near term (assuming that other requirements for actionability, such as salience and legitimacy, have also been met).*

This text clarifies that we are not discounting salience and legitimacy.

Even here, I would have really liked the authors to examine the role of values and tradeoffs in defining credibility, especially recognizing that scientific uncertainty can often be weaponized to delay action.

We agree that scientific uncertainty can be, and has been, weaponized to delay action on climate change. But we think it is important to distinguish between claims which are sufficiently accepted to justify near-term action and those which are not. Our case study illustrates some problems that can arise from a standard that is too weak. We think a high-confidence threshold would be too strict, in part because of the danger JB notes. In proposing a medium-confidence threshold, we are trying to find a workable middle ground.

Determining if a study has multiple lines of evidence to support it is, as we can see from the comments, subjective and the authors have not demonstrated that their definition improves decision making.

Minimizing subjectively (i.e., maximizing objectivity) in decision-making is one of our core goals. It should be uncontroversial that greater objectivity will improve decision-making, even if perfect objectivity is an ideal that can only be approached asymptotically.

If science is social knowledge, as argued by Longino (1990), then objectivity is maximized when new ideas are examined critically by a diverse group of experts, as in the IPCC process. Objectivity can be compromised, on the other hand, if the unit of knowledge is taken to be a single peer-reviewed paper, or a report written by a small group of scientists who do not provide detailed, traceable accounts of their reasoning. The DP16 case study shows what can happen when scientific claims are communicated to practitioners without a vetting process designed to maximize objectivity. We think our criterion would have improved decision-making if it had been

applied to recent sea-level adaptation guidance in California and elsewhere. We acknowledge that a more general demonstration of our criterion's fitness for decision-making is beyond the scope of a short article.

As the current paper reads, it seems more like it is specifically designed as a refutal of DeConto and Pollard's (2016) projections and the impact they have had as a high-end scenario in planning and adaptation decisions rather than as a general framework.  If that is the goal then the study should, perhaps be framed as such.

Our aims are both specific and general, with the case study of DP16 offering support for a general framework. To acknowledge the role of this case study in supporting our arguments, we modified the last sentence in the first paragraph of the Recommendations as follows (ll. 277–278, with new text in italics): "This criterion is informed by IPCC practices, by philosophical arguments that scientific knowledge is social knowledge, *and by the DP16 case study*."

To give some additional examples, the explosive disintegration of the Larsen B ice shelf and subsequent acceleration of tributary was not anticipated by any models nor was the sudden acceleration and retreat of Jakobshavn, Pine Island or Thwaites Glacier.  The disintegration of Conger ice shelf was also a surprise, although perhaps shouldn't have been. Going beyond glaciology, the pre-eminent physicists of the time Lord Kelvin famously said that "Heavier than air flying machines are impossible" a mere 8 years before the first airplane successfully flew (Shoemaker, 1995).  The Harvard Economic Society announced that "A severe depression like that of 1920-1921 is outside the range of probability" on November 16, 1929 (Shoemaker, 1995).  I could go on with other examples in which real world events failed to adhere to the academic consensus.  Here the point isn't that scientists can be wrong (of course we are!), but that how communities, stakeholders and decision makers choose to incorporate information depends heavily on values, resources and objectives.  Would we be better off stress testing, strategizing or planning to deal with low probability, high impact events? I don't know and I would be loath for scientists to insist on being on the arbiter of these value laden decisions.  Actionable doesn't always mean infrastructure and it might mean longer term strategic thinking to be better positioned to incorporate new information as it becomes available. Perhaps one useful framing for the authors is that their approach might be most suited to mega infrastructure investments that are expensive and take decades to plan and build. Different criteria would be appropriate for different scales of intervention.

We generally agree with these statements. In the revised manuscript, we endorse stress-testing and strategizing within DMDU frameworks as an appropriate way to deal with low-confidence, high-impact processes. Also, we clarify that we use the term "actionable" to refer to near-term action, including large infrastructure investments. We agree that longer-term strategic thinking can incorporate low-confidence information—appropriately differentiated from information that meets our criteria—and that different criteria are appropriate for different physical scales and time scales of intervention.

We also agree that "how communities, stakeholders and decision makers choose to incorporate information depends heavily on values, resources and objectives". At the same time, it is not impinging on the values of stakeholders and decision-makers to tell them that certain scientific claims are characterized by low confidence or deep uncertainty. For example, practitioners are routinely told not to use temperature projections from a single GCM in planning. Similarly, we are suggesting that practitioners should not use projections from a single paper without confirmation from the broader scientific community.

My penultimate point illustrates the degree of mischief that is possible if terminology and definitions aren't tightened up and illustrated with multiple case studies.  The authors argue that DeConto and Pollard's study of sea level rise isn't credible (actionable in their words) because there isn't multiple lines of evidence to support ice cliff instability.  I suspect that the paleo-record results in more ambiguity than the authors concede here,  but nonetheless this is a fair point. However, we can also say that there is zero evidence that the calving front will remain stationary and very little evidence to support *any* calving law used in ice sheet modeling.  By the authors same logic, we might conclude that *none* of the projections of sea level rise are credible (actionable in the authors words).  We can apply this same reasoning to other processes. Should we doubt the entire enterprise of climate modeling because of the quasi-empirical treatment of clouds, precipitation and aerosols?  Clearly, this is not what the authors intend, but it might be an unintentional side effect if not clarified.  To this end, the IPCC, NOAA, NASA Sea level team and multiple organizations are involved in summarizing literature and play a well documented role in determining "credibility" and these organizations form one step in a chain of boundary organizations.  Here there needs to be dialogue with the role played by these existing structures and organizations and the interplay with other boundary organizations.

Our argument does not imply that "none of the projections of sea level rise are credible". In particular, we view the IPCC medium-confidence sea-level projections as credible enough to be actionable, if other criteria for actionability are also met. The same would be true of other projections associated with medium (or high) confidence.

We have worked to tighten and clarify our terminology and definitions. Additional case studies are beyond the scope of this work, but we agree that case studies from other fields would be valuable. We agree about the importance of dialogue, as emphasized in the Recommendations.

My final point relates to the definition of "action".  As noted by Knaggard (2014), one of the common actions that stakeholders take is to decide that the scientific uncertainty is currently too large and devote resources to research to better quantify and hopefully reduce uncertainties. By this definition, the action associated with high impact/low probability impacts might be more research (potentially coupled with adaptive decision making).
However, the most common action that decision makers take, however, is simply to do what is currently politically and technically feasible (Knaggard, 2014).  These decisions, based on political expediency and co-benefits depend more on the solution space than scientific uncertainty and so establishing credibility is less important than salience and legitimacy.

We agree that adaptive decision making and additional research are appropriate ways to deal with the possibility of high impact/low probability events. We have clarified these points in the revised manuscript, for example in the discussion of DMDU. We also agree that in some cases, credibility may be less central than salience and legitimacy.

I'm very sympathetic to the authors goals, but I'm not sure that, to use the authors definition, their definition of "actionable" has multiple, independent lines of evidence to support its use and adoption. I think it would be hard to demonstrate this in the limited space available, but I think it would be a great addition if the authors built their case through a larger number of case studies and literature review and applied their own criteria of multiple lines of evidence to their own proposed definition of actionable science.

While we have clarified what we mean by "actionable", our main goal is not to define every aspect of actionable science. Rather, we have proposed a practical criterion for identifying scientific claims that are sufficiently accepted to be actionable. To support this criterion, we have offered multiple lines of argument and evidence drawn from IPCC practice, the philosophy of science, and the scientific case study. But we would be the first to say that our article is not the last word on the subject.

**Reply to Judy Lawrence, Marjolijn Haasnoot, and Robert Lempert**

We are researchers and science brokers who work with SLR science in the context of decision making under uncertainty (DMDU). We recognize the challenge of dealing with multiple scenarios that are frequently updated as new understanding emerges. However, the proposal set out by LBM runs the risk that we wait until there is certainty before taking adaptation action. This would also not meet the precautionary test in the UNFCCC which underpins the Paris Agreement, and lead to adaptation decision delay.

We thank these researchers for their thoughtful comments and also for their contributions to the DMDU literature, which have been helpful in sharpening our ideas. Some omissions and lack of clarity in our initial submission have created the appearance of disagreement, when in fact we support the use of DMDU methods in adaptation planning.

We suggest that medium confidence is a lower and more appropriate standard for adaptation action than "certainty". Our goal is not to delay adaptation decisions, but rather to help practitioners take robust near-term actions with limited information, including deep uncertainty. We think that if a particular action (e.g., building costly infrastructure to protect against 3 m of SLR by 2100) is likely to be unnecessary, then it is consistent with the precautionary principle to delay action and wait for more information. At the same time, it is prudent to make dynamic plans that could trigger such actions in the future, if justified by new evidence.

There are approaches that have emerged and assessed in IPCC AR6 for considering the dilemma that decision makers find themselves in as new science continues to emerge and decisions have to be made under uncertainty over the trajectory and pace of change. These DMDU approaches and methods are able to deal with large uncertainties and updated (climate)

information. The LBM paper mentions adaptive planning and integrating SLR information into planning but does not elaborate what this means or how these approaches can address the problem of over and under investment in adaptation. This omission leads to a very directive solution that would not be decision relevant (salient nor legitimate).

We regret that our initial proposals came across as too directive. While we want to reduce overinvestment in adaptation, we also don't want to push practitioners toward underinvestment. We agree that DMDU methods are good tools for dealing with large uncertainties and updated information. The revised manuscript describes how DMDU can be used to reduce the risks of both overinvestment and underinvestment.

Furthermore, we agree with Bassis's comment that the LBM paper only focuses on the matter of credibility and that salience and legitimacy are critical for decision making under uncertainty to enable local context to inform decisions. The response from LBM that the paper is not about **how** the science is used, misses the point that science does not sit in a vacuum outside how it may be used. In fact the authors have defined their problem with an example of how the projections were 'misused' in the San Francisco example. Science and its use are inextricably linked.

As stated in the reply to JB, we agree that salience and legitimacy are important for decision making, although they are not the focus of our article.

We agree that science and its use are inextricably linked. We did not mean to imply that we are unconcerned with how science is used. To the contrary, our goal is to ensure that the use of science in adaptation planning is consistent with the evidence supporting that science.

One cannot set a fixed criteria/standard for a changing context (both science and societal values-which decision makers weigh up). The question of the standard of the science cannot be divorced from a discussion about how sea-level rise projections are used. Using DMDU methods such as Dynamic Adaptive Pathways Planning (DAPP; Haasnoot et al. 2019) or Robust Decision Making (Lempert et al. 2019) to stress test adaptation options against a range of scenarios gives decision makers an idea of how sensitive each strategy is and then this information can be weighed against the other considerations that the decisionmakers must take into account as representatives of their communities, now and in the future. Given that surprises around polar ice processes and feedback are occurring, even high-end/low confidence scenarios are useful to remind decision makers that one cannot rule out such outcomes, as also stated in IPCC AR6. A DAPP approach enables adaptation to be broken down into near-term actions and mid-to long-term options to avoid lock-in of investments that are costly to adjust in the future and which shift the adaptation costs to future generations. It helps to prepare and keep options open or create them through innovation and planning, allowing further adaptation if necessary. DAPP thus enables more robust decisions to be made and contingency actions to be ready.

We support the use of DMDU methods such as DAPP and RDM, with accompanying stress tests and sensitivity studies, and we agree that it is essential to separate near-term actions from mid- to long-term options. We hope our revisions have clarified this agreement.

We would also note that most communities planning adaptation action do not use DMDU approaches today. We think that communities without the experience or resources to use these more advanced tools could benefit from the approach we propose.

Given the time it takes for long lived infrastructure projects to be designed and implemented, especially for coastal adaptation, to high and rapid sea level rise, a precautionary approach has merit. Not considering such scenarios can run the risk of being too late or invest in the wrong measures resulting in high sunk costs and transfer costs. Considering them on the other hand gives decision makers greater confidence in a changing situation (science and societal values).

The nature of the decision process is lightly addressed in the paper beyond one example. A range of approaches as to how SLR projections can be used should be proffered. These can be found in the literature.

For example, in the Netherlands a group of experts (the Signal Group) advise the government on relevant new research which is then assessed for its potential implication and need for further research or actions (Haasnoot et al. 2018). The follow-up research from Deltares and KNMI led to further assessment on the need to reassess the adaptation strategy. It also raised awareness of the long-term higher sea levels, even if they were in low global warming scenarios, and that the current adaptive plan would not be sufficient, and that transformative adaptation would be needed. While near-term actions were not changed immediately, it was recognised that preparations were needed to be able to further adapt, if such an accelerated SLR became a reality.

In New Zealand, revised national coastal hazards and climate change guidance (Ministry for the Environment 2024) has adopted a tailored approach for different types of decisions, using DAPP to stress test adaptation options with downscaled global scenarios that enable polar ice responses and vertical land movement to be incorporated into SLR which vary around the coast. Periodic revisions are undertaken to reflect new science and a precautionary approach is adopted considering at least a 100 year timeframe to account for change and uncertainty. Specific infrastructure guidance is also available (Lawrence and Allison, 2024).

Adaptive pathways planning and monitoring for signals is also done in practice in the Thames Estuary plan (Environment Agency, 2012; Ranger et al 2013) and New York (Blake et al., 2019; Rozenzweig). They also plan regular ongoing assessment and evaluation of changes (e.g. ongoing and every 6 six years in New York and in the Netherlands).

Thank you for sharing these examples. We agree that it is helpful to give examples of approaches (including DMDU) that have worked well. We added the following paragraph at the end of Section 4 (ll. 261–272):

Finally, we would like to give two examples of practitioner guidance that avoided the pitfalls described above. First, the most recent sea-level guidance from the Met Office Hadley Centre (Palmer et al. 2018) retained high-end projections adopted in 2009. Citing DP16, the authors noted that "marine ice cliff instability has been proposed as an important potential feedback" but added that "further research is required to strengthen the observational evidence for, and prevalence of, this mechanism". Second, the 2017 New Zealand coastal adaptation guidance (Ministry for the Environment, 2017) proposed a high-end ("H+") scenario of 1.05 m GMSL by 2100, based on AR5, as part of a dynamic adaptive pathways planning (DAPP) strategy. The updated guidance (Ministry for the Environment, 2024) retained the DAPP approach and used the AR6 medium-confidence projections for SSP5-8.5 to design an H+ scenario with 1.1 m GMSL by 2100. The new projections were described as a "plausible upper range" for SLR and were recommended for "high-end stress testing of adaptation options and pathways". The AR6 low-confidence projections were assigned a limited role for "further stress testing" related to long-lived coastal development and managed-retreat options. Since the core recommendations in the New Zealand guidance have not relied on low-confidence science, there has been no whiplash.

Again, we appreciate the many thoughtful comments from the community. We conclude our response with one note. None of the above comments suggests a boundary between scientific claims that are accepted or credible enough for near-term adaptation action and those that are not. Perhaps the authors of these comments think that all peer-reviewed claims (or claims from some subset of journals) are actionable, but they do not offer epistemic arguments for a peer-review criterion. We continue to believe that adaptation planning would benefit from differentiating between actionable science and science that is peer-reviewed but not actionable. Without this differentiation, we have seen in recent years and believe we will continue to see confusion, whiplash, and adaptation planning inefficiencies.

**References**

Cash, D. W., Clark, W. C., Alcock, F., Dickson, N. M., Eckley, N., Guston, D. H., Jaäger, J., and Mitchell, R. B.: Knowledge systems for sustainable development, PNAS, 100, 8086–8091, https://doi.org/10.1073/pnas.1231332100, 2003.

DeConto, R. M. and Pollard, D.: Contribution of Antarctica to past and future sea-level rise, Nature, 531, 591–597, https://doi.org/10.1038/nature17145, 2016.

Lempert, R. J., Popper, S. W., and Bankes, S. C.: Shaping the Next One Hundred Years: New Methods for Quantitative, Long-Term Policy Analysis, RAND Corporation, 2003.

Longino, H. E.: Science as Social Knowledge: Values and Objectivity in Scientific Inquiry, Princeton University Press, Princeton, NJ, ISBN0-691-02151-5, 262 pp., 1990.

Mastrandrea, M. D., Mach, K. J., Plattner, G. K. *et al.*: The IPCC AR5 guidance note on consistent treatment of uncertainties: a common approach across the working groups, Climatic Change, 108, 675–691, https://doi.org/10.1007/s10584-011-0178-6, 2011.

Ministry for the Environment: Coastal hazards and climate change: Guidance for local government, Wellington: Ministry for the Environment. Pub. 1341, https://environment.govt.nz/assets/publications/Files/coastal-hazards-guide-final.pdf, 2017.

Ministry for the Environment: Coastal hazards and climate change guidance, Wellington: Ministry for the Environment. Pub. 1805, https://environment.govt.nz/assets/publications/Coastal-hazards-and-climate-change-guidance-2024-ME-185.pdf, 2024.

O'Neill, B. C., Tebaldi, C., van Vuuren, D. P., Eyring, V., Friedlingstein, P., Hurtt, G., Knutti, R., Kriegler, E., Lamarque, J.-F., Lowe, J., Meehl, G. A., Moss, R., Riahi, K., and Sanderson, B. M.: The Scenario Model Intercomparison Project (ScenarioMIP) for CMIP6, Geosci. Model Dev., 9, 3461–3482, https://doi.org/10.5194/gmd-9-3461-2016, 2016.

O'Neill, B.C., Carter, T.R., Ebi, K. et al.: Achievements and needs for the climate change scenario framework, Nat. Clim. Change, 10, 1074–1084, https://doi.org/10.1038/s41558-020-00952-0, 2020.

Palmer, M., Howard, T., Tinker, J., Lowe, J., Bricheno, L., Calvert, D., Edwards, T., Gregory, J., Harris, G., Krijnen, J., Pickering, M., Roberts, C., and Wolf, J.: UKCP18 Marine Report, Met Office, UK, https://ukclimateprojections.metoffice.gov.uk, 2018.

---

## Author Response (AR2)

**Reply to Rebecca Priestley**

We thank the reviewer for another set of constructive comments. Our responses are below, in blue font.
* * *
I enjoyed reading this revised draft of the paper. The issues around clarity of language that I identified in the first draft have been resolved, and the paper is now clear and focused in its message.

I just have some small issues that, if resolved, will make the paper even more impactful and useful.

1. Looking at your definitions of practitioner and policymaker in the footnote on page 1, I'm wondering where 'policy advisers' fit? Under your definition would they count as practitioners? In New Zealand the policy adviser works for a government agency and "gathers evidence, analyses policy issues, develops policy options and prepares policy advice for defined policy issues" (I took this from an online job ad). I've not looked into how they are defined in other countries (or how much this job description is even used) but It seems to me that policy advisers are a key audience for this paper - but I don't think they'd define themselves as practitioners. If they are to be included in the word practitioners, perhaps that could be made clear?

We agree that policy advisors are a key audience. We think it is more straightforward to group them with practitioners than to create a separate category, so we modified the footnote as follows (new text in italics):

> … Practitioners include staff charged with evaluating and developing solutions to climate-related risks who work for local, state/provincial, and national governments, land managers, corporations, and other public or private sector entities. *This term would also include policy advisors—those who analyze complex issues and develop options, given a defined policy.* We distinguish practitioners from policymakers—the legislators and other government officials who create laws and regulations.

2. In line 40, inside the brackets, I suggest the semicolon is replaced by a colon, or some other thought given to punctuation and structure to make it clear that DMDU is an abbreviation of the last four words, and is separate to the citation (it took me a while to figure out!)

Sorry for the confusion. We modified the sentence:

> Researchers and practitioners have developed frameworks for decision making under deep uncertainty (DMDU) (Marchau et al., 2019a).

3. Lines 59-61, from 'would say' to 'long run' raised questions for me (if you're drawing on philosophy of science there's a lot to unpack with the statement 'science can be objective' and the next sentence lacked clarity) and I wondered if these lines were even necessary – the next paragraph is very clear and sets out the key points that Longino is making. I would suggest that the first two paragraphs of section 2 be revised to cut much of lines 59-61.

We cut much of lines 59–61 and combined the first two paragraphs of Section 2 into a single paragraph:

> … Furthermore, many philosophers of science—including Helen Longino, from whose work we draw here—have argued that the organization of scientists in communities leads to greater understanding and more reliable predictions. Longino (1990) emphasizes …

4. On line 85, the phrase 'their attention may be drawn' was passive in a way the rest of the paper is not – perhaps this could be rephrased as 'they are likely to be exposed to' or something similar (just a suggestion)

We switched to active voice:

> When practitioners learn about climate research through media reports, they are likely to give undue attention to a small number of studies in high-impact journals …

5. On line 89, I suggest that 'If they rely on media accounts for the' is changed to 'If they rely on media accounts to alert them to' – otherwise it suggests they're getting their science from the media reports, rather than from the papers the media reports are covering.

That's a good point. We changed the sentence as suggested.

6. On line 280, is 'vetted' the right word? This word is fine in its meaning of 'evaluation or appraisal' but it can also imply this is done for some kind of 'official approval or acceptance' which might go further than intended by this statement?

We meant "vetted" in the sense of evaluation and appraisal. To avoid confusion, we replaced "vetted" with "evaluated".

7. Finally, a key message in this paper, set out in the abstract, and on page 4, and in the recommendations, is that science should only be considered actionable if 'evaluated by a diverse group of scientific experts'. It's made clear that this happens in the IPCC system, but do you have any other examples of where this happens? Or if it currently only happens in IPCC reports, how or where else might it happen? This doesn't need to be fully set out in the paper but I think the paper should make it clear that either (a) we need new systems and structures to do this, or (b) there are already many examples (other than IPCC) of where this is done. Or is

this paper a specific call for IPCC advice to be the only science taken into account for sea level rise decision making? If so, this could be stated explicitly.

Thank you for the suggestion. While the IPCC reports are not the only place this happens, there are relatively few other examples (at least for sea-level projections), so we think new structures would be valuable. We added the following sentence at the end of the second paragraph of the Recommendations:

> Since IPCC reports are infrequent, we recommend new structures that regularly bring together scientific experts to assess ongoing research on sea-level rise and other fast-evolving topics.

The last paragraph refers to the aim of PEERS to "create practice-centered collaboration with a diverse group of scientific experts". This would be one such structure.